# Tetraspanin 4 stabilizes membrane swellings and facilitates their maturation into migrasomes

Raviv Dharan [1,2,7], Yuwei Huang [3,7], Sudheer Kumar Cheppali [1,2], Shahar Goren [1,2,4], Petr Shendrik [1], Weisi Wang [3], Jiamei Qiao [3], Michael M. Kozlov [2,5], Li Yu [6] & Raya Sorkin [1,2] ✉

Migrasomes are newly discovered cell organelles forming by local swelling of retraction fibers. The migrasome formation critically depends on tetraspanin proteins present in the retraction fiber membranes and is modulated by the membrane tension and bending rigidity. It remained unknown how and in which time sequence these factors are involved in migrasome nucleation, growth, and stabilization, and what are the possible intermediate stages of migrasome biogenesis. Here using live cell imaging and a biomimetic system for migrasomes and retraction fibers, we reveal that migrasome formation is a two-stage process. At the first stage, which in biomimetic system is mediated by membrane tension, local swellings largely devoid of tetraspanin 4 form on the retraction fibers. At the second stage, tetraspanin 4 molecules migrate toward and onto these swellings, which grow up to several microns in size and transform into migrasomes. This tetraspanin 4 recruitment to the swellings is essential for migrasome growth and stabilization. Based on these findings we propose that the major role of tetraspanin proteins is in stabilizing the migrasome structure, while the migrasome nucleation and initial growth stages can be driven by membrane mechanical stresses.

Cells modulate their shape in striking ways during vital tasks such as movement, division and vesicle trafficking[1,2]. A particular case of membrane shaping is the formation of the recently discovered signaling organelle termed migrasome. Migrasomes form by local swelling of retraction fibers, the cylindrical protrusions of cell membranes that form as a result of cell migration along external substrates[3,4]. Migrasomes can grow up to several micrometers in diameter[5], and allow cells to release contents such as chemokines at specific locations, hence, transmitting signals to surrounding cells through the relevant chemokine receptors. Recently, evidence emerged showing that

migrasomes play essential roles in several fundamental cellular processes. A physiological role of migrasomes has been recently demonstrated in vivo: organ morphogenesis in zebrafish was found to depend on migrasomes that contain signaling molecules and provide regional biochemical cues that allow correct cell positioning[6]. Migrasomes contain mRNA and proteins that can transfer into recipient cells and functionally modify them[7]. Migrasomes have also been found to accumulate damaged mitochondria, and dispose them to the extracellular environment, providing cells the ability to maintain the quality of the mitochondrial pool[8]. Migrasomes are also involved in the

[1]School of Chemistry, Raymond & Beverly Sackler Faculty of Exact Sciences, Tel Aviv University, Tel Aviv, Israel. [2]Center for Physics and Chemistry of Living Systems, Tel Aviv University, Tel Aviv, Israel. [3]School of Basic Medical Sciences, Xi'an Jiaotong University, 710049 Xi'an, China. [4]School of Mechanical Engineering, The Ivy and Aladar Fleischman Faculty of Engineering, Tel Aviv University, Tel Aviv, Israel. [5]Department of Physiology and Pharmacology, Sackler Faculty of Medicine, Tel Aviv University, Tel Aviv, Israel. [6]The State Key Laboratory of Membrane Biology, Tsinghua University-Peking University Joint Centre for Life Sciences, Beijing Frontier Research Center for Biological Structure, School of Life Sciences, Tsinghua University, 100084 Beijing, China. [7]These authors contributed equally: Raviv Dharan, Yuwei Huang. ✉e-mail: rsorkin@tauex.tau.ac.il

pathogenesis of brain injury after cerebral ischemia in humans, characterizing them as potential therapeutic target in acute ischemic stroke[9]. After formation and maturation, migrasomes separate from the cell body. In other words, migrasomes are organelles that become extracellular vesicles (EVs) once they detach from the retraction fibers' membrane. Thus, the detached migrasomes are new members of the family of EVs, which are important mediators of cell–cell communication[10], as well as spreading of disease[11,12], including cancer metastasis[13]. Migrasomes can also potentially be used for diagnostic and therapeutic purposes once their biological roles and formation mechanisms are characterized and better understood.

Paramount factors in migrasome formation are tetraspanin proteins (TSPAN). The migrasomes and the retraction fibers are enriched with TSPANs[14]. TSPAN family, including 33 known members in humans, are small proteins all with four transmembrane domains present in every cell type[15,16]. They are involved in various physiological processes including cell adhesion, migration, immune response, fusion and signaling in diverse organs[17], yet their mechanism of action is still poorly understood.

TSPANs can interact with themselves and with other integral proteins, lipids and adhesion molecules, forming a distinct class of membrane domains[15,18–20], which are thought to be closely related to their functionality. Previously, it was established that both TSPAN and cholesterol are necessary for migrasome formation, which involves formation of domains in the retraction fiber and migrasome membranes[14]. Further evidence showed that over-expression of 14 different TSPANs enhanced migrasome formation. Among them, tetraspanin 4 (TSPAN4) was found to be the most effective[14]. Along with the crucial function of TSPANs in migrasome formation, it was further theoretically suggested that biophysical parameters such as membrane tension and elasticity are contributing to migrasome formation[14]. The time sequence in which each of these factors is involved in the evolution of migrasomes and the possible intermediate stages of this process are still unknown.

Here, we discover that the process of migrasome formation consists of two stages. At the first stage, membrane swellings form along retraction fibers with relatively small amount of TSPAN4 on them. At the second stage, TSPAN4 molecules move toward and onto these swellings, which is accompanied by the swelling's growth to several microns size and their transformation into migrasomes. We demonstrate this mechanism by imaging migrasome generation in live cells and validate it by recreating the conditions leading to migrasome-like vesicle generation in a biomimetic model system. Our results suggest that migrasome nucleation and initial growth can be driven by membrane mechanical stresses, while the stabilization of these structures occurs upon TSPAN recruitment.

## Result and discussion

### Migrasome biogenesis occurs in a two-stage mechanism

We followed the formation of migrasomes of Normal rat kidney (NRK) cells over-expressing TSPAN4-GFP stained with FM4-64 membrane dye under confocal microscopy.

Migrasomes formed along retraction fibers following cell migration (Fig. 1a). Closer to the cell body, FM4-64 enriched puncta could be seen along the retraction fibers (Fig. 1a and Supplementary Movie 1). Further from the cell, large migrasomes were observed, enriched with both TSPAN4-GFP and FM4-64 (Fig.1a and Supplementary Movie 1). The fact that mature migrasomes were enriched with TSPAN4 while the newly generated ones were not, suggests the initial stage of migrasome biogenesis to occur ahead of the TSPAN4 recruitment. To further test this hypothesis, we conducted time-lapse imaging of living cells by using structural illumination microscopy (SIM) (Fig. 1b). At an early stage, FM4-64 was locally enriched as puncta on the retraction fibers (Figs. S1 and S2a). The small puncta could move along the retraction fibers and coalesce to bulge out

from the thin retraction fibers as small swellings. At this initial stage, TSPAN4-GFP was relatively homogeneously distributed along the retraction fibers. At the next stage TSPAN4-GFP was gradually recruited onto the swellings, which grew into migrasomes (Fig. 1b, c and S1 and S2b). To address this process quantitatively, we followed the formation of 252 individual migrasomes. We divided the migrasomes into four groups based on the lipid and TSPAN4-GFP fluorescence intensity at the initial stage of migrasome formation versus the migrasome growth stage (detailed examples of these stages are shown in Fig. S1). The groups are defined as follows: (1) initial-stage is red (enriched in FM4-64), growth stage is yellow (enriched in FM4-64 and TSPAN4-GFP), (2) initial stage is red, growth stage is red, (3) initial stage is green (enriched in TSPAN4-GFP), growth stage is green, (4) initial stage is yellow, growth stage is yellow (Fig. 1d). Most of the mature migrasomes (almost 80% (Fig. 1d)) started to form from FM4-64 enriched small puncta and then grew in size concomitantly with further TSPAN4-GFP enrichment.

### TSPAN4 enrichment is essential for migrasome growth and stabilization

We have further concluded that TSPAN4 recruitment to the migrasomes is necessary for migrasome stabilization. We observed two populations of migrasomes (Fig. 1e). The population indicated by the white arrows initially increased in size, but eventually shrunk back within the time course of the experiment. The migrasomes indicated by yellow arrows grew and stably maintained their large sizes throughout the experiment (Fig. 1e). The main visible difference between these two types of migrasomes was in their TSPAN4 signal, which was increased in the stable migrasomes (Fig. S3). Overall, these results strongly suggest a two-stage mechanism of migrasome biogenesis, the first stage being formation of FM4-64 positive small swellings, and the second stage being migrasome growth and stabilization through TSPAN4 recruitment.

In order to further confirm our observations, we compared NRK-TSPAN4-GFP cells to WT NRK cells (Fig. S4a and Supplementary Movies 2 and 3). We found that while endogenous TSPAN4 levels are sufficient for migrasome stabilization, TSPAN4 overexpression increased the numbers of pre-migrasomes, which matured into stable migrasomes. (Fig. S4b). This result demonstrates the stabilization effect of TSPAN4. TSPAN4 also enhanced the formation of pre-migrasomes, as can be seen from comparison between the total number of pre-migrasomes in the NRK-TSPAN4-GFP cells and that for WT cells (Fig S4c). A probable reason is a TSPAN4-driven increase of the number and length of the retraction fibers[21]. Overall, the number of mature migrasomes increased significantly due to TSPAN4 over-expression (Fig. S4d).

Next, we wanted to test whether TSPAN4 expressed at endogenous levels can stabilize migrasomes. We followed migrasome biogenesis in WT MGC803 and in MGC803 TSPAN4-KO cells (Fig. S4e and Supplementary Movies 4 and 5). The deletion of TSPAN4 significantly increased the percentages of pre-migrasomes, which dissipated during the experiments (Fig. S4f). The number of mature migrasomes in the WT MGC803 cells was significantly higher than in TSPAN-KO cells (Fig. S4g), demonstrating that endogenous TSPAN4 levels are sufficient for stabilization of pre-migrasomes and stimulation of their growth. Furthermore, as observed in the NRK cells experiments, TSPAN4 increased the amount of pre-migrasomes (Fig. S4h). Our results show that, while TSPAN4 contributes indirectly to the formation of pre-migrasomes by contributing to the formation of larger numbers of the retraction fibers and by increasing the retraction fiber length, as described previously[21], it does not participate in the formation of initial swellings appearing as small puncta with relatively low TSPAN4 concentration. At the next stage, however, TSPAN4 is recruited to the swellings, stabilizes them, and facilitates their growth into mature migrasomes.

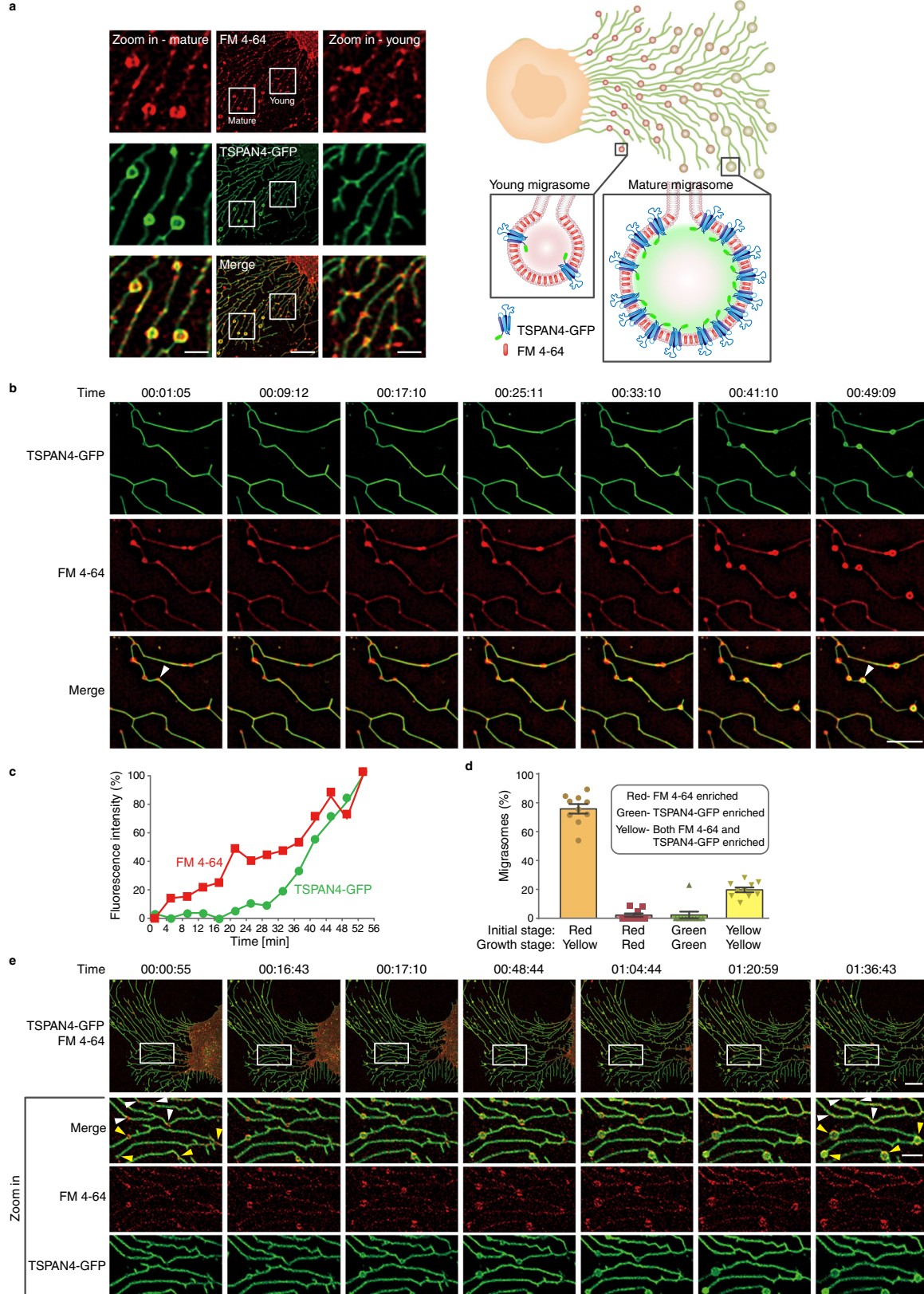

## Tension jump induces swelling formation on the membrane tube

In order to get insight into the main factors underlying the two-stage mechanism of migrasome biogenesis, we designed a biomimetic system emulating a cell with a retraction fiber and enabling an experimental simulation of migrasome formation. We used optical tweezers combined with confocal fluorescence microscopy and micropipette aspiration[22,23]. As a model for the cell, we used a giant plasma-membrane vesicle (GPMV)[24], which we generated from HEK293T or NRK cells expressing TSPAN4-GFP (Fig. 2 and S5). To imitate a retraction fiber, we pulled a membrane tube out of an aspirated GPMV by attaching a polystyrene bead to the vesicle and then moving the bead

**Fig. 1 | Two-stage mechanism of migrasome formation. a** Confocal images of NRK TSPAN4-GFP (green) cells stained by FM4-64 (red). Scale bar, 10 µm; zoom in, 2.5 µm. The experiment was repeated independently three times with similar results. On the right, schematic representation of young migrasomes with low TSPAN4-GFP concentration and mature migrasomes with TSPAN4-GFP enrichment. **b** Time-lapse images of NRK TSPAN4-GFP cells stained by FM4-64. Imaging by structural illumination microscopy (SIM). Scale bar, 5 µm. **c** Normalized fluorescence intensity as function of the time of TSPAN4-GFP and FM4-64 on a representative migrasome in **b**, indicated by a white arrow. Normalization based on

retraction fiber fluorescence. **d** Statistical analysis of four different groups of migrasomes during biogenesis (red–yellow, red–red, green–green, yellow–yellow), based on a series of time-lapse images of NRK TSPAN4-GFP cells stained by FM4-64 under confocal microscopy. $N = 252$, from 10 individual cells from three independent experiments. Error bars are standard error of the mean (SEM). **e** Confocal Time-lapse images of NRK TSPAN4-GFP cells stained by FM4-64. White arrows point to migrasomes that form and shrink back; yellow arrow heads point to growing migrasomes. Scale bar, 10 µm; zoom in, 3 µm. Time in **b** and **e** is hh:mm:ss. Source data are provided as a Source Data file.

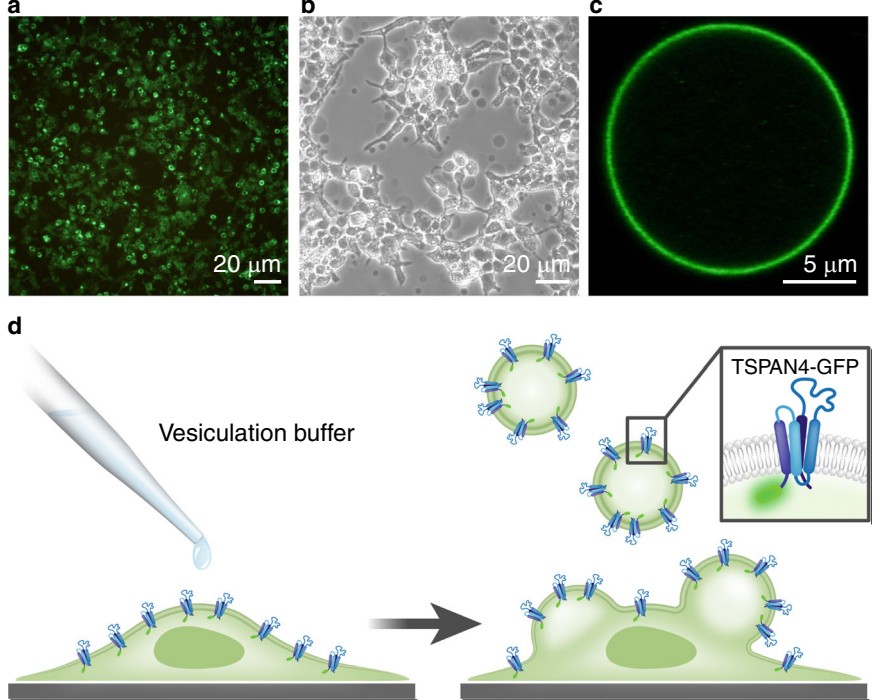

**Fig. 2 | Formation of giant plasma-membrane- vesicles (GPMVs) from transfected HEK293T cells expressing TSPAN4-GFP. a** Fluorescence microscopy image of HEK293T cells expressing TSPAN4-GFP 24 h after transfection. **b** Phase contrast microscopy image of HEK293T cells, expressing TSPAN4-GFP, after treatment with a vesiculation buffer. GPMVs, which appear dark in phase contrast image, can be seen floating in the sample or attached to the cells. **c** Confocal

microscopy image of a GPMV containing TSPAN4-GFP. **d** Schematic representation of a transfected cell, treated with vesiculation buffer, producing GPMVs with TSPAN4-GFP in their membrane. Illustration adapted from[28]. The experimental procedure shown in **a–c** was repeated in seven independent experiments (for HEK293T-TSPAN4-GPMVs).

away by optical tweezers (Fig. S6). This setup enabled us to control the membrane tension of the GPMV (and, hence, the tube diameter through setting the aspiration pressure), to measure the force pulling the tube by the optical tweezers, to perform confocal fluorescence imaging of the system, and monitor the system evolution in real time via bright-field microscopy.

To simulate the initial stage of the migrasome formation we reasoned that the generic factor driving local swellings of membrane tubules might be an abrupt increase of membrane tension, which is known to lead to tube pearling instability[25]. A crucial role of membrane tension in migrasome formation is supported by the previous work[14]. To test this idea, we designed an experiment of a two-step tension application to a membrane tube. First, the tube was pulled out of a GPMV subject to a relatively low membrane tension such that the tube radius was relatively large (Fig. 3a and S7). Next, we rapidly increased the GPMV aspiration pressure and, hence the membrane tension[26] (Supplementary Movie 6). The time of the tension increase was substantially shorter than the time needed for the tube relaxation to a new equilibrium configuration of a homogeneous cylinder with a reduced cross-sectional radius corresponding to the new level of the tension, which required a slow decrease of the intra-tubular volume through

liquid flow into the GPMV. This condition of a transiently constant volume corresponded to that of the pearling instability[25]. Indeed, the abrupt increase of tension led to generation of the migrasome-like local swellings of the tube (Fig. 3a; S7 and S8). The swellings were formed at random locations along the tube (Fig. S9) and were able to move along the tube (Fig. S10). The constricted regions of the tube were enriched with TSPAN4 whereas the swellings were enriched with DiI-C12 (Fig. 3b). These results were consistent for GPMVs derived from HEK293T and NRK cells. The average threshold of tension jump, which induced periling instability, was different for the two types of GPMVs, as NRK-GPMVs required significantly higher tension jump for swelling formation (Fig. S11 and methods). This increase in the threshold tension value might result from a higher bending rigidity of NRK cells, which have higher endogenous concentration of tetraspanins than HEK293T cells (as tetraspanins are known to increase the membrane bending rigidity[14]).

## TSPAN4 proteins assemble into domains

As a following step, we examined whether TSPAN4 in our biomimetic system self-organizes into clusters. We found that while being highly enriched in the tubular membranes, TSPAN4 can be ununiformly

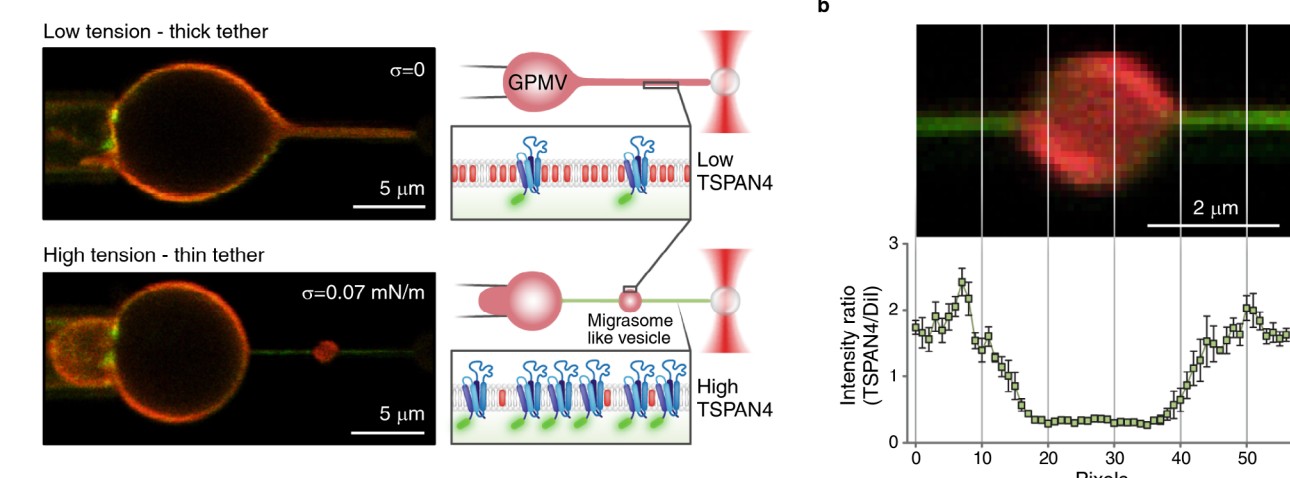

**Fig. 3 | Swelling formation on a membrane tube induced by rapid tension increase. a** Confocal microscopy images of a membrane tube pulled from a GPMV containing TSPAN4-GFP (green) and DiI-C12 (red) aspirated with a micropipette. In the top image, the suction pressure was zero (corresponds to zero tension applied, $\sigma = 0$). Next, the tension increased immediately to 0.07 mN/m. On the right side, schematic representation of tension-induced swelling formation assay.

**b** TSPAN4-GFP and DiI-C12 fluorescence intensity ratio of membrane tubes containing a swelling ($n = 4$ membrane tubes, having swelling with relatively same size, pulled from 4 vesicles from 3 independent experiments, error bars are SEM). On the top, a representative image of a membrane tube containing a swelling. Source data are provided as a Source Data file.

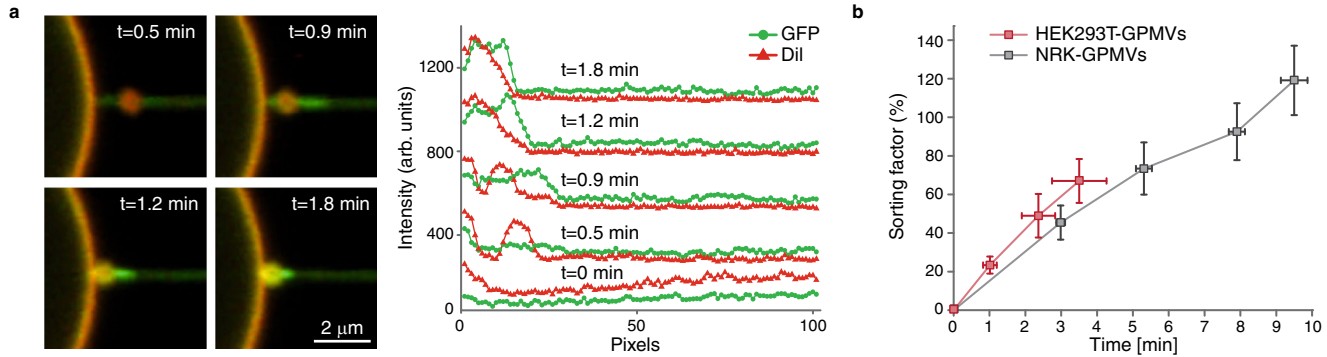

**Fig. 4 | TSPAN4 domains recruitment to the tubule swellings. a** Time-lapse confocal microscopy images of a membrane tube pulled from a GPMV containing TSPAN4-GFP (green) and DiI-C12 (red). Following swelling formation, TSPAN4 clustering followed by TSPAN4 swelling enrichment was observed. On the right, the fluorescence intensity of DiI-C12 and TSPAN4-GFP along the tube and the swelling at the indicated times ($t = 0$ correspond to the intensity of the tube just before the swelling formation). **b** Time dependence of the percentage change of

TSPAN4 sorting factor, $S = \frac{(I_{TSPN4-GFP}/I_{DiI-C12})_{bubble}}{(I_{TSPN4-GFP}/I_{DiI-C12})_{vesicle}}$, which is the fluorescence intensity ratio of GFP and DiI-C12 on the swelling compared to the vesicle. Red line corresponds to HEK293T-GPMVs containing TSPAN4-GFP ($n = 8$ membrane tubes pulled from 7 vesicles from 5 independent experiments, error bars are SEM) and black line corresponds to NRK-GPMVs containing TSPAN4-GFP ($n = 9$ membrane tubes pulled from 6 vesicles from 4 independent experiments, error bars are SEM). Source data are provided as a Source Data file.

distributed but can also form mobile puncta along the tubes (Fig. S12a). According to the previous observations in cellular retraction fibers[14], these puncta can be classified as clusters. We further demonstrated that TSPAN4 clusters formed also in flat membranes of GPMVs exposed to shear forces, which were induced by buffer flow (Fig. S12b). In this experiment, the GPMVs were injected into a microfluidics chamber under high pressure (1.5 bar), which led to substantial shear forces acting on the vesicle membranes. Altogether our results demonstrate the tendency of TSPAN4 to cluster.

### Recruitment of TSPAN4 domains to the swellings

Next, we sought to examine whether our model system is able to recreate TSPAN enrichment in the swellings, as observed for the migrasome biogenesis in live cells. For HEK293T-GPMVs, In 8 out of 25 swellings, we observed TSPAN4-GFP migration to the swellings prior to tube rupture (Fig. 4a). TSPAN4 migrated to the swellings in the form of TSPAN4-enriched domains and stayed on the swellings until the

tube ruptured. To quantify the swelling enrichment with TSPAN4, we calculated the relative intensities of TSPAN4-GFP and DiI-C12 on the swellings compared to the GMPV, which increased with time (Fig. 4b). Furthermore, TSPAN4 containing tubes exhibited an augmented tendency to rupture as compared to the control tubules, i.e., without TSPAN4 (Fig. S13a). The swellings were observed to remain intact after the rupture (Fig. S13b). The likely reason for the lack of partitioning of TSPAN4 domains to the swellings in all the experiments was the tubule rupture, which did not leave enough time for the domains to migrate. The rupture may be promoted by the membrane structural defects emerging along the boundaries of the TSPAN4-enriched domains, as suggested by previous reports showing that the lipid phase separation can lead to the tubule rupture[27]. The rupture of the membrane tube occurred less frequently in tubes pulled from NRK-GPMVs (Fig. S14a). This enabled us to follow TSPAN4 enrichment in the swelling over longer times. In 9 out of 10 swellings, we saw a significant increase in TSPAN4 concentration (Fig. 4b and S14b, c).

We suggest the following explanation for TSPAN4 recruitment to migrasomes. While TSPAN4 molecules exhibit high-positive intrinsic curvature corresponding to an effective molecular shape of an inverted cone[28,29], TSPAN4 assembly into clusters and larger domains can reduce its intrinsic curvature. This leads to migration of the domains onto the membrane swellings that have a smaller curvature and, therefore, a better curvature compatibility with the domains. The hypothesis of lower intrinsic curvature of large TSPAN-enriched domains compared with single TSPAN proteins is further supported by the finding that TSPANs associate with cholesterol[20]. Moreover, we quantified the cholesterol enrichment in migrasomes, and found that cholesterol levels increased significantly in mature migrasomes (concomitantly with TSPAN4 enrichment), supporting this hypothesis (Fig. S15). Domains that consist of TSPAN associated with cholesterol and other lipids and proteins should have an intrinsic curvature substantially lower than that of individual TSPAN molecules, the latter having particularly high intrinsic curvature[28].

### Stabilization of swellings by TSPAN4

Finally, we tested whether our model system exhibits slower dissipation of membrane swellings in the presence of TSPAN4. We conducted control experiments with GPMVs that did not contain over-expressed TSPAN4. Strikingly, for both HEK293T-GPMVs and NRK-GPMVs the swellings demonstrated a behavior very similar to that for the TSPAN4-depleted migrasomes in live cells, where they dissipated in the absence of TSPAN4 (Fig. 5b). Specifically, after an abrupt tension increase in GMPVs lacking TSPAN4, the swellings formed along the membrane tube, similarly to the results presented in Fig. 3, and then rapidly disappeared within an average timespan of 29 s after formation for HEK293T-GPMVs (Fig. S13c) and 73 s for NRK-GPMVs (Fig. S14d). In the

experiments with GMPVs containing TSPAN4, however, the formed swellings were much more stable (Fig. 5A), their lifetime being at least six times longer than that measured in the control experiments for HEK293T-GPMVs (Fig. S13c) and at least four times longer for NRK-GPMVs (Fig. S14d). The increase in the swelling's lifetime in WT NRK-GPMVs compared to WT HEK293T-GPMVs is most likely due to the significantly higher endogenous TSPANs expression levels of NRK cells[14].

Overall, we investigated the temporal pathway of migrasome formation in conjunction with TSPAN4 dynamics on retraction fibers of live cells and designed a biomimetic system emulating these processes in a minimal artificial system. In this system, although we only followed the initial swelling process, we recreated the crucial aspects of migrasome biogenesis observed in live cells: formation on membrane tubes of initial migrasome-like local swellings having relatively low TSPAN concentrations, formation of TSPAN domains and their recruitment to the swellings, dissipation of the swellings in the absence and their stabilization in the presence of TSPAN. The differences between the migrasome formation in cells and the biomimetic system can be related to cellular factors such as the cytoskeleton, which may modulate the process in live cells[5,9,30,31] but are absent from the artificial system. For example, membrane-buckling induced by cytoskeleton was previously implicated in vesicle budding[32,33], and regulators of actin dynamics were shown to result in microvesicle formation in tumor cells[34]. In addition, cortical cytoskeleton may modulate membrane tension[35–37]. Based on the obtained results, it can be concluded that the migrasome formation proceeds in two sequential steps: formation of local swellings on the tubular retraction fibers, and stabilization of these swellings by TSPAN-based membrane domains. Our results suggest that the migrasome biogenesis can be driven by a very

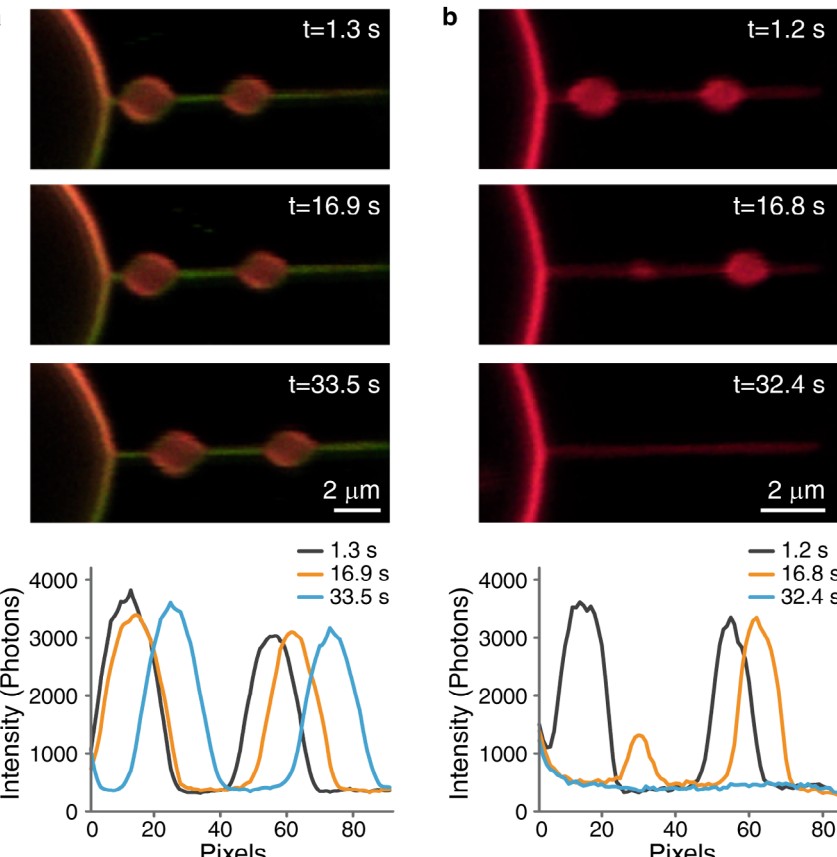

**Fig. 5 | TSPAN4 inhibits swelling dissipation.** Time-lapse confocal microscopy images of membrane tubes with swellings, dyed with DiI-C12 (red) in the presence of (**a**) and without (**b**) TSPAN4-GFP (green), following tension jump of $\triangle\sigma = (5.0 \pm 0.1) \times 10^{-5}\,\text{N/m}$. At the bottom: fluorescence intensity profiles of DiI-C12 along the membrane tube and swellings at the indicated times.

limited set of factors: the first step driven by membrane tension with possible involvement of additional factors in live cell membranes, and the second step controlled by specific proteins of the TSPAN family. The particular cellular mechanisms by which membrane tension and TSPAN-based clusters shape the migrasomes, while being partially addressed in the previous work[14], await substantial elaboration. Our results illuminate the mechanism of migrasome biogenesis and demonstrate the crucial role of TSPAN proteins in membrane shaping processes.

## Methods

### Cell culture

HEK293T (ATCC CRL-3216TM) and Normal rat kidney (NRK, ATCC CRL-6509TM) cells were cultured at 37 °C and 5% CO$_2$ in DMEM (Gibco-Thermo Fisher scientific 11995065) supplemented with 10% Fetal bovine serum (biological industries, 04-001-1A) and 1% penicillin–streptomycin (Gibco-Thermo Fisher scientific 15140122). NRK cells were gifted by Natalie Elia lab in BGU, Israel.

MGC803 gastric carcinoma cells (RRID:CVCL 5334, ICLAC-00588) were gifted from Zhijie Chang lab from Tsinghua. MGC803 and NRK cells were cultured at 37 °C and 5% CO$_2$ in DMEM (Gibco, C11995500BT) supplemented with 10% Fetal bovine serum (Vivicum, 9906-85) and 1% penicillin–streptomycin (Cienry, CR15140).

### TSPAN4 expression plasmids, cell transfection, and giant plasma-membrane vesicles (GPMVs) isolation

Complementary DNAs of mouse TSPAN4 were cloned into pEGFP-N1. HEK293T or NRK cells were plated at 20% confluency in 25 cm$^2$ flask (Romical) coated with poly-L-lysine (Sigma-Aldrich P6282) to keep the cells attached during the blebbing process and to minimize cell debris in solution. At 50% confluency, cells were transiently transfected with 5 µg DNA using Lipofectamine 2000 (Invitrogen, Thermo Fisher scientific) according to the manufacture's protocols and then grown 24 h for protein expression. GPMVs were produced according to a published protocol[38]. Briefly, following TSPAN4-GFP expression the cell membrane was stained (except from experiments shown in Figs. S10 and S12) with DiI-C12 membrane dye (Invitrogen, Thermo Fisher scientific D383), washed with GPMV buffer (10 mM HEPES, 150 mM NaCl, 2 mM CaCl2, pH 7.4) twice, and incubated with 1 mL of GPMV buffer containing 1.9 mM DTT (Sigma-Aldrich 1019777701) and 27.6 mM formaldehyde (Sigma-Aldrich F8775). Secreted GPMVs were then collected and isolated from the cells and immediately used for optical trapping experiments. Among the GPMVs, about 10% contained TSPAN4-GFP in their membranes. In order to verify the presence of TSPAN4-GFP in the GPMVs' membrane we scanned the GPMVs using a 488 nm laser before each measurement. For the control experiments GPMVs that did not contain TSPAN4-GFP in their membrane were used.

### Cell imaging

**Cell sample preparation.** NRK cells with stable TSPAN4-GFP expression (NRK-TSPAN4-GFP), WT NRK, WT MGC803 or MGC803-TSPAN4-KO cells were seeded (around $4 \times 10^3$ cells) into 3.5 cm glass-bottom confocal dish (Cellvis D35C4-20-1.5-N), which was pre-coated with 10 µg/mL fibronectin (Gibco, Thermo Fisher scientific PHE0023) and grew for 12–15 h. The cells were stained with 5 µg/mL FM4-64 (Invitrogen, Thermo Fisher scientific T3166) for 15 min at 37 °C.

**Confocal microscopy imaging.** Cell imaging shown in Fig. 1a, e and S1 was conducted under galvanometer scanning mode using a NIKON A1 confocal microscope fitted with a 100× oil objective. The laser power was 0.5% for 488 nm and 3% for 561 nm and each field of 1024 × 1024 pixels was imaged. For time-lapse imaging, the interval was 4 min, and the duration was at least 5 h. Cell imaging shown in Figure S4 was conducted by a Nikon HD25 microscope with 60x objective and each

field of 1024 × 1024 pixels was imaged for at least 6 h with 5 min interval. The laser power was 3% for 561 nm, 54.35 µW is the maximal laser power.

**SIM imaging.** The cells shown in Fig. 1b were imaged by structured illumination microscopy (Nikon N-SIM S) with 4 min interval and 1.5 h duration, and then reconstructed by a standard stack-reconstruction process.

### Quantitative analysis of migrasomes

For the analysis shown in Fig. 1d, 252 migrasomes of 11 NRK-TSAPN4-GFP cells from 4 movies conducted at three independent experiments were analyzed, each movie 4 h in length, with 60 frames in total. Fully formed migrasomes were chosen in the last frames of the movies, and then tracked back to their initial stage in the earlier frames. Only migrasomes that formed in the time course of the movie were included in the analysis. For the growing stage, several frames were included in the analysis in order to validate the presence of fully formed migrasomes due to possible changes in the focus. An example of initial stage, starting from red puncta, and growing stage is shown in Supplementary Fig. S1.

For the analysis shown in Fig. S4b–d, 1391 pre-migrasomes of 54 WT NRK cells from 16 movies of four independent experiments, and 1899 pre-migrasomes of 55 NRK-TSPAN4-GFP cells from 14 movies of four independent experiments were analyzed.

For the analysis shown in Fig. S4f–h, 724 per-migrasomes of 31 WT MGC803 cells from 11 movies of four independent experiments, and 274 pre-migrasomes of 30 MGC803-TSPAN4-KO cells from 13 movies of four independent experiments were analyzed.

### Proteolysis and mass spectrometry analysis

In order to confirm TSPAN4 overexpression in the GPMVs, we conducted mass spectrometry (MS) measurements, which showed that TSPAN4 abundance in GPMVs generated by TSPAN4-transfected cells increased 1562-fold compared to control GPMVs (GPMVs that were generated from WT HEK293T cells as described above). The samples were extracted in 10 mM DTT, 100 mM Tris and 5% SDS, boiled in 95 °C for 10 min and sonicated twice for 10 min. The samples were precipitated in 80% acetone overnight and washed three times with 80% acetone. The protein pellets were dissolved in 8.5 M Urea (Sigma-Aldrich U6504) and 400 mM ammonium bicarbonate (Sigma-Aldrich 09830). The protein amount was estimated using Bradford readings. Proteins were reduced with 10 mM DTT (Sigma-Aldrich D9163) at 60 °C for 30 min, modified with 40 mM iodoacetamide (Sigma-Aldrich I6125) in 100 mM ammonium bicarbonate (room temperature for 30 min in the dark) and digested in 1.5 M Urea, 66 mM ammonium bicarbonate with modified trypsin (Promega), overnight at 37 °C in a 1:50 (M/M) enzyme-to-substrate ratio. An additional trypsin digestion was performed for 4 h at 37 °C in a 1:100 (M/M) enzyme-to-substrate ratio. The resulting tryptic peptides were desalted using C18 stage tips (homemade, 3 M company, USA) dried and resuspended in 0.1% Formic acid (Fluka analytical 94318-250 ML-F). The peptides were resolved by reverse-phase chromatography on 0.075 × 300-mm fused silica capillaries (J&W) packed with Reprosil reversed phase material (Dr. Maisch GmbH, Germany). The peptides mixture was eluted with a 5 to 28% of linear gradient of solvent B (95% acetonitrile (J.T. Baker 9821)) with 0.1% formic acid (Thermo Fisher scientific 85178) in water for 60 min followed by a 28 to 95% linear gradient for 15 min in solvent B at flow rate of 0.15 µL/min. MS was performed by Q Exactive Plus mass spectrometer (Thermo Fisher scientific) in a positive mode using repetitively full MS scan followed by high-collision dissociation (HCD) of the 10 most dominant ions selected from the first MS scan. The MS data was analyzed using Proteome Discoverer 2.4 software with Sequest (Thermo-Fisher scientific) search algorithm against Human Uniprot database with 1% FDR. Semi quantitation was done by calculating the

peak area of each peptide based its extracted ion currents (XICs), and the area of the protein is the average of the three most intense peptides from each protein.

## Tube pulling from aspirated GPMVs

The experiments were performed using a C-trap® confocal fluorescence optical tweezers setup (LUMICKS, Amsterdam, the Netherlands) made of an inverted microscope based on a water-immersion objective (NA 1.2) together with a condenser top lens placed above the flow cell. The optical traps are generated by splitting a 10 W 1064-nm laser into two orthogonally polarized, independently steerable optical traps. To steer the two traps, one coarse-positioning piezo stepper mirror and one accurate piezo mirror were used. Optical traps were used to capture polystyrene microbeads. The displacement of the trapped beads from the center of the trap was measured and converted into a force signal by back-focal plane interferometry of the condenser lens using two position-sensitive detectors. The samples were illuminated by a bright-field 850-nm LED and imaged in transmission onto a metal-oxide semiconductor (CMOS) camera. Confocal fluorescence microscopy: The C-Trap uses a 3 color, fiber-coupled laser with wavelengths 488, 561 and 638 nm for fluorescence excitation. Scanning was done using a fast tip/tilt piezo mirror. For confocal detection, the emitted fluorescence was descanned, separated from the excitation by a dichroic mirror, and filtered using emission filters (Blue: 500–550 nm, Green: 575–625 nm and Red: 650–750 nm). Photons were counted using fiber-coupled single-photon counting modules. The multimode fibers serve as pinholes providing background rejection. For confocal imaging the 488 nm and 532 nm lasers were used for GFP and DiI-C12 excitation with 5% and 1% laser power, respectively, 54.34 μW is the maximal laser power, and the emission detected in three channels (Blue, Green, Red).

**Experimental chamber.** Polydimethylsiloxane (PDMS) walls were placed on the bottom cover slip (Thorlabs CG15KH1) and mounted onto an automated XY-stage. The GPMVs sample was added to the chamber and after about 15 min, a few drops of oil were put on the sample surface to prevent evaporation. A micropipette aspiration setup including micromanipulator (Sensapex) holding a micropipette with diameter of 5 μm (BioMedical instruments) connected to a Fluigent EZ-25 pump was integrated into our optical tweezers instrument. Before each experiment, the zero-suction pressure was found by aspirating a 3.43 μm polystyrene bead (Spherotech) into the pipette and reducing the suction pressure until the bead stopped moving. A membrane tube was pulled from aspirated GPMVs using beads trapped by the optical tweezers. First, a membrane tube was pulled at relatively low suction pressure (0.05–0.1 mbar, correspond to $1.2–2 \times 10^{-5}$ N/m membrane tension), then the suction pressure was reduced to zero (corresponds to zero applied membrane tension) for about 15 s. Then, we increased instantaneously the suction pressure to values in the range of 0.2–0.8 mbar (correspond to $2–15 \times 10^{-5}$ N/m membrane tension) for HEK293T-GPMVs or 0.5–1.1 mbar (correspond to $10–25 \times 10^{-5}$ N/m membrane tension). The control experiments were conducted on HEK293T-GPMVs and NRK-GPMVs (which did not contain TSPAN4-GFP). The same tension jumps used for TSPAN4-GPMVs were done on the GPMVs from the control group ($\pm 5 \times 10^{-6}$ N/m). Microfluidics: To induce shear forces on TSPAN4-GPMVs, the GPMVs were injected at 1.5 bar into a 5-channel laminar flow cell (LUMICKS, Amsterdam, the Netherlands), which was on the C-trap® confocal fluorescence optical tweezers stage. The pressure was reduced to zero and following the settlement of the GPMVs at the bottom of the flow cell, they were scanned using a 488 nm laser at 5% laser power.

## Data analysis

Data acquisition was carried out using Bluelake, a commercial software from Lumicks. This software stores experimental data acquired with the C-trap in HDF5 files, which can be processed using Lumicks' Pylake python package. Images of the confocal scans were reconstituted from photon count per pixel data in the HDF5 files using Pylake. All data analysis was performed with custom-written Python scripts. Fluorescence intensity profiles were obtained from the images by averaging the photon count of the relevant fluorescent channel (Blue or Green) in the region of interest.

## Reporting summary

Further information on research design is available in the Nature Portfolio Reporting Summary linked to this article.

## Data availability

The mass spectrometry proteomics data have been deposited to the ProteomeXchange Consortium via the PRIDE partner repository with the dataset identifier PXD039664. The map of the vector containing TSPAN4-GFP was deposited in OSF repository[39]: https://osf.io/ze3g8/?view_only=0774f3b44784404f91e8d6ff45664721. All other data supporting the findings of this study are available in this article, its supplementary information, or the Source Data file. Source data are provided with this paper.

## Code availability

All codes used for data analysis are available at https://gitlab.com/raviv_dharan/protein-curvature-sensitivity-analysis.

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

## Acknowledgements

R.S. acknowledges support by the ISRAEL SCIENCE FOUNDATION (grant No. 1289/20). S.K.C. acknowledges support by the Ratner Center for Single Molecule Science. MMK was supported by Deutsche Forschungsgemeinschaft (DFG) through SFB 958 "Scaffolding of Membranes", and Israel Science Foundation grant 3292/19, and holds Joseph Klafter Chair in Biophysics. Y.H. acknowledges support by the National Natural Science Foundation of China (32070691). We acknowledge the Smoler Proteomics Center at the Technion for the mass spectrometry analysis, and Dafna Antes for assistance with graphics and illustrations.

## Author contributions

R.D. and Y.H. contributed to the manuscript equally. R.D., Y.H., L.Y., M.K., and R.S. designed research; R.D., Y.H., S.G., P.S., W.W., J.Q., R.S. performed research; R.D., Y.H., W.W., J.Q. analyzed data; S.K.C. contributed to sample preparation; R.D., Y.H., M.M.K., L.Y., and R.S. wrote the paper.

## Competing interests

The authors declare no competing interests.
