## [Peer Review File · Nature Communications]

Tetraspanin 4 stabilizes membrane swellings and facilitates their maturation into migrasomesREVIEWER COMMENTS

Reviewer #1 (Remarks to the Author):

In the manuscript entitled “Tetraspanin 4 mediates migrasome formation via a two-stage mechanism”, Dharan et al. investigated the mechanisms behind the formation of an organelle called migrasome, known to be formed by local swelling of retraction fibers formed at the trailing edges of migrating cells. The authors proposed a two-step mechanism based on changes in membrane tension followed by recruitment of TSPAN proteins to migrasome. Overall, I believe the manuscript is very well written and the results are extremely interesting and well organized. My comments are listed below:

1- The manuscript abstract (first 2 paragraphs) has much more than 150 words. The authors also did not present an introduction section. Furthermore, from the 3rd paragraph onwards results/discussion are presented. It would be interesting if the authors could rewrite the abstract, could create an introduction section and could also delimit each of the sections on the manuscript text. As an extra suggestion, authors should create subsections under results/discussion, but I will leave that decision to the editorial board and authors.

2- In the legend of figure 1D, please specify whether the results are represented as mean +- SD or mean +-SEM.

3- In Figures 1 and 1E zoom in, there appears to be a decrease in TSPAN4 fluorescence in retraction fibers around the forming migrasomes. However, this reduction does not seem to extend over several micrometers. Is the decrease in fluorescence due to the movement of proteins to the migrasome region? Or is it already a possible effect of the decrease in retraction fiber diameter due to the increase in membrane tension? The authors could address this by staining the membrane with a more general fluorophore and measuring the retraction fiber diameter in the region close to the migrasome. The new result would help a lot with the subsequent correlations.

4- No supplementary movies 1 and 2 were added, which made it difficult to interpret some of the dynamic results described.

5- Lines 66-67: Using all the movies and results already obtained by the authors, would it be possible to characterize the mean time for the event to occur? This quantitative information could be very useful.

6- Figure 3: Did the authors perform the OT-micropipette experiment using tension values other than 0.07nN/m? How did the authors select this 0.07nN/m? Is there a threshold value of tension for the event to occur in the biomimetic system?

7- Lines 185-188 and Lines 191-194: the authors used the results for the biomimetic assays to conjecture possible explanations for the phenomenon occurring in cells. I believe it is totally valid but it needs to be much more discussed. For example, the authors never mentioned the cytoskeletal elements present in retraction fibers and how could they participate in the process. It is already known that changes in cell membrane tension are dependent on the cytoskeleton as well as membrane composition. As the results for the biomimetic assays seem to partially resemble the migrasomes but within different time scales (seconds for the GPMV and minutes for the migrasomes), could it be that the presence of cytoskeleton in retraction fibers influenced, somehow, the tension threshold and the time scale? This could be better discussed by the authors and would help to strengthen the mechanism proposed.

Reviewer #2 (Remarks to the Author):

This paper from the Sorkin lab describes experiments aimed at characterising the cellular mechanisms through which migrasomes – a relatively recently-described category of cell-derived vesicles - bud-off from retraction fibres. The authors have previously shown that a tetraspannin, TSPAN4 is key to migrasome formation and here, they use biophysical approaches to further interrogate the mechanisms through which TSP4 and membrane tension conspire to drive migrasome budding. The authors propose a sequential mechanism through which increased membrane tension drives initial migrasome budding from retraction fibres. These buds are then subsequently stabilised – perhaps to facilitate their fission – by recruitment of TSPAN4 to these nascent buds.

In general, the study is elegant and well-executed and provides evidence of a possible mechanism for migrasome genesis. The biophysical approaches are good ones to use to interrogate this process, and the experiments are properly conducted and controlled. Nevertheless, I feel that in order to properly establish that TSPAN4 functions in the way that the authors claim – i.e. to stabilise nascent migrasomes – the authors need to conduct further experiments and controls.

Major points:

1. My main point is that the authors need to determine to what extent the mechanism and phenomena that they describe rely on overexpression of TSPAN4 and the extent to which this tetraspannin can mediate stabilisation of migrasome-like buds on retraction fibres when TSPAN4 is present at endogenous levels. The experiments presented in Fig 1E show nicely that there is a recruitment of GFP-TSPAN4 to nascent migrasomes correlates with their stabilisation. The authors also show that migrasome-like structures which form on the membrane tubes pulled from GMPVs made from GFP-TSPAN4 overexpressing cells are 5-times more stable than those formed from GMPVs made from cells

that don't overexpress GFP-TSPAN (Fig. 5). However, to establish that such a mechanism could be biologically pertinent, the authors should address the following questions:

- a) what is the level of endogenous TSPAN4 in the cells used for this study? Is this insufficient to support migrasome stabilisation?
- b) Can the authors perform experiments (similar to those in Figs 1E and Fig. 5) in cell line(s) which endogenously express high levels of TSPAN4? Such as the MGC-803 line used for their previous study.
- c) Is there a correlation between endogenous TSPAN4 expression and swelling stabilisation?
- d) Does knockdown/CRISPR of endogenous TSPAN4 oppose nascent migrasome swelling/stabilisation? Could the authors perform the GPMV/optical tweezers experiments using the wt and TSPAN4 KO MGC-803 lines used in the previous study to investigate the role of endogenous TSPAN4 on migrasome swelling and stabilisation.

I feel that answering these questions is particularly important because the authors' previous study in NCB (2019) has already established that a role for TSPAN4 and biomechanics in migrasome production and used GPMVs – amongst other approaches – to do this. The present manuscript extends these findings in a way that is interesting but, because the present study relies on TSPAN overexpression, the relevance of sequential mechanism proposed here requires a demonstration of whether this can be supported by TSPAN4 when endogenously expressed.

2. In the example shown in Fig. 4, the nascent migrasome forming from the GPMVs appears to form on the tubule at some distance (about 1 micron) from the GPMV. However, the recruitment of GFP-TSPAN4 to the structure seems only occur when it has moved very close (apparently touching) to the GPMV. Does this always occur like this? It would help if the authors could determine this using the gallery of movies that they have collected and determine the distance from the GPMV that initial swelling and GFP-TSPAN4 recruitment respectively occur.

3. I do not understand the purpose of the proteomics as conducted in this study? It currently seems to be used to determine that GFP-TSPAN4 is present in GPMVs. I feel that it is more important to determine which TSPANs are 'sorted' into GPMVs and how enriched these are.

4. I do not seem to be able to find the movies for this paper on the NC manuscript tracking site. It is not possible to properly assess these parts of the paper without seeing these movies.

Reviewer #3 (Remarks to the Author):

The authors present a series of findings regarding the temporal sequence of events resulting in the formation of migrasomes. While further study of migrasomes is certainly of interest to the EV community, this study would benefit from additional work.

Comments:

1. The experimentation in the manuscript relies on 2 cell lines: 1 used in live cell imaging and another used for their GPMV studies. To gauge the impact of the results a more thorough analysis of key observations across multiple cell types is necessary. The authors themselves refer to this on lines 124/125.
2. The reliance on overexpression studies is concerning, particularly in light of the data presented in Figure 5. How frequent or relevant are supposed migrasomes if in the absence of overexpression their formation is significantly reduced?
3. Is the formation of migrasomes in their live cell imaging model substrate and/or adhesion dependent? Tetraspanins are known to associate with integrin receptors for example.
4. The manuscript does not adequately place the work within the broader context of EV biology. How, for example, does their mechanism of formation relate to the previously published works examining the role of hydrostatic pressure in the formation of EVs. Are migrasomes released, can they be isolated, are their contents altered with TSPAN4 overexpression?
5. What percentage of migrasomes mature? Is TSPAN4 being presented as a definitive marker of migrasome biogenesis, or are other FM labeled vesicular structures also migrasomes simply without TSPAN4?
6. Are more migrasomes formed in cells that migrate more readily? Is this linked to TSPAN4 expression in those cell types?
7. The reader is left confused regarding the Dil-C12 experiments described in line 119. Is this meant to be disordered lipid phase rather than liquid phase? The data seems somewhat contradictory to what the authors propose regarding the role of cholesterol.
8. The section on TSPAN4 clustering is confusing and should be reworked. The authors utilize the same model as the paper being referenced and yet simultaneously state that the TSPAN4 expression is uniformly distributed but also mobile puncta.
9. Methods are incomplete. Microfluidic work, for example, is not included. There is also no relevant statistical analysis conducted on the data.

Reviewer #4 (Remarks to the Author):

In this paper the authors wanted to explore the formation kinetics and stabilization dynamics of migrasomes; with a focus on tetraspanin 4 (TSPN4). For this study, the authors utilized two systems: live cell confocal microscopy (NKT Cells) and a biomimetic system using giant plasma membrane vesicles (GPMVs). For both experimental investigations, they developed cell-lines over expressing TSPN4. In the biomimetic model, they simulated migrasome-like vesicles generation and used innovative biophysical imaging techniques to evaluate TSPN4 recruitment to membrane swellings that anticipate migrasome formation.

The study seeks to demonstrate that TSPN4 mediates migrasome formation in a two-step process, starting with membrane swelling along retraction fibers and subsequent TSPN4 migration into the swelling site for migrasome growth and stabilization. Their overall conclusion of the two-step formation process is a novel finding. The topic matter is important and investigations such as this are needed to shed light into the mechanisms that leads to the formation of these novel extracellular signaling structures. The authors of this work have led some of the earliest work into migrasomes and their expertise comes across.

The strengths of the paper include novel experimental models that allow for the pursuit of combined optical data alongside dynamic tension changes in their biomimetic system (confocal optical tweezers combined with their model GPMVs). The live cell imaging experiments are more standard, yet the authors do an excellent job in both obtaining excellent images while also translating those images into quantitative data. The use of the GPMVs as a biomimetic model is clever and provides a mechanism (albeit a bit artificial) to explore TSPN4 recruitment dynamics in microsome formation.

The manuscript also had areas of weakness. The conclusions drawn felt overly broad considering that they only using one cell line in each model system, with few replicates. The authors failed at times to help the reader understand some of their experimental decisions (why did they switch the cell line between experiments?), while provided limited background on some of their more innovative methods. As an example, the reasoning behind why the authors chose the focus on TSPN4 is never clearly stated. On line 125 it is hinted it was from previous work from the group, albeit this is not done in a clear way. Overall, this is an interesting piece of work paving the way for the understanding of recently discovered cell signaling structures. With some improvements, this work can have a major impact on the study of environmental cues involved in cellular signaling and communication.

Major corrections:

1. While the work is of high interest, it is also weakened by the lack of continuity between the two models. The authors should help to justify why HEK cells were used for the biomimetic model and not the normal rat kidney cells used for the live imaging work. Further, when making conclusions about the overall process of migrasome formation (as the authors do on line 189), these conclusions should be affirmed in more than one cell line (per experimental methods).

The number of replicates for each experiments is also unclear. Currently, their findings are interesting, but only affirmed in this one set up that they have created.

2. Their conclusions are also weakened by the limited number of replicates for their experiments. The live cell confocal microscopy conclusions come from four videos. Are they all from the same plating of cells? Are they across different dates? What variations did they include to make sure that these results aren't specific to just one preparation of cells? These details should be provided. Similarly, they should also be provided for the optical tweezers experiments.

3. Limited details are provided on the statistical methods used. Some error bars are shown without reference to what they mean (Fig. 1D). In other times, "SEM" is mentioned, without details.

4. Hypotheses are made and at times conclusions drawn through reference to their past work (role of cholesterol in TSPAN orientation/formation; lines 153-159) and this falls flat in this work. Data should be collected and details provided in this manuscript should be provided or the authors should refrain from the speculation. For example, could the authors stain cholesterol on these models to see if it associated with TSPN4?

5. To access and understand both the biology and technology, the reader is required to do a significant background reading. While brevity is always needed, additional details need to be provided. The introduction could have additional background on the biology of migrasomes formation (what is known / not known), while also explaining what previous work led to their choice of TSPN4 and to support their choice of these particular experiments.

6. The formatting for this work appears off. There is no abstract provided. There is no Discussion. Overall, the manuscript lacks structure. It reads as though it was prepared for another journal, but then sent to Nature Communications.

7. Lines: 85-86 – Relating to Fig 1E, needs a graph showing the TSPN4 signal quantifying these visible findings.

8. Related to not providing enough detail for how this work contributes to the field, the references are lacking with just 17 provided. While the field is new, this feels like a lot of previous work was left unreferenced. Charrin et al (2003) work about the interactions of tetraspanins with cholesterol was never mentioned and could support some claims made in the manuscript (p.e).

Minor corrections:

Methods overall: Important details are missing from the methods section and one cannot replicate the work as presented. What type of fibronectin is used (human, bovine?). What type of serum? Suppliers are frequently not stated and sometimes they are stated without the region/country, but further in the text this information is provided. Needs to be consistent. Every reagent should at least have a supplier name and then should be written as supplier name/region/country. Suppliers name are the minimum information required.

Some specific notes:

207 - Suppliers for cell lines.

230 - should come at 211 – Easier to know how cells that are being imaged were made.

212 - Density? Supplier for fibronectin FM-4 64?

226 - How many frames?

233 - Density?

235 - Empty vector control?

238 - for most of experiments? What about the others?

247 - brought? I don't understand what this means

251-256 - sentence is too long. Proteins "were" reduced?

260-261 - wording confusing. "Linear gradients of 5% to 28% for 60 minutes, 28% to 95% for 15 minutes...."

262 - Thermo is not a company. Thermo Fisher Scientific.

264 - MS – Full name followed by abbreviation is missing in text – mass spectrometry (MS). Abbreviations need to be consistent throughout the manuscript and be next to the first time a word is mentioned.

305 – LUMICKS was referenced before without country. Be consistent.

Notes on figures

Figure 1 – This reviewer suggests that the reader would benefit from having 'C' and 'D' side by side below figure B. Also, C and D are lacking statistical reference.

47- indicated in B by a white arrow

48 - kinds=groups? Kinds sounds non specific

Figure 2 B - why not fluorescent as well?

150 - Published, under review, in preparation? Ref?

151 - This needs a reference. I think they are referencing their previous work (1) but something needs to be referenced here.

Figure 4. B – No stats

Figure 5. Maybe label A and B makes it easier to follow?

Supplementary figures

- Inconsistent TSPN4 nomenclature

- Figure 2 – lest=left

- Needs attention to the legends.

Response to reviewers of manuscript NCOMMS-22-08791-T "Tetraspanin 4 mediates migrasome formation via a two-stage mechanism"

We thank the editor for handling our manuscript and for appreciating the importance of revealing the mechanism for migrasome formation. We are also thankful to the reviewers for their careful reading of our manuscript and the constructive comments. Below we address all the comments in full detail. Reviewers' comments are in blue, our responses are in black, and revisions incorporated in the manuscript are highlighted by yellow background.

Response to 1st reviewer:

In the manuscript entitled "Tetraspanin 4 mediates migrasome formation via a two-stage mechanism", Dharan et al. investigated the mechanisms behind the formation of an organelle called migrasome, known to be formed by local swelling of retraction fibers formed at the trailing edges of migrating cells. The authors proposed a two-step mechanism based on changes in membrane tension followed by recruitment of TSPAN proteins to migrasome. Overall, I believe the manuscript is very well written and the results are extremely interesting and well organized. My comments are listed below:

1- The manuscript abstract (first 2 paragraphs) has much more than 150 words. The authors also did not present an introduction section. Furthermore, from the 3rd paragraph onwards results/discussion are presented. It would be interesting if the authors could rewrite the abstract, could create an introduction section and could also delimit each of the sections on the manuscript text. As an extra suggestion, authors should create subsections under results/discussion, but I will leave that decision to the editorial board and authors.

We thank the reviewer for the positive remarks and the helpful suggestions. We rearranged the manuscript and created an Introduction section as well as subsections of the Results section.

2- In the legend of figure 1D, please specify whether the results are represented as mean \pm SD or mean \pm SEM.

We thank the reviewer for pointing out this issue. In figure 1D, the results were presented as mean \pm SEM. We have added this information to our revised manuscript.

3- In Figures 1 and 1E zoom in, there appears to be a decrease in TSPAN4 fluorescence in retraction fibers around the forming migrasomes. However, this reduction does not seem to extend over several micrometers. Is the decrease in fluorescence due to the movement of proteins to the migrasome region? Or is it already a possible effect of the decrease in retraction fiber diameter due to the increase in membrane tension? The authors could address this by staining the membrane with a more general fluorophore and measuring the retraction fiber diameter in the region close to the migrasome. The new result would help a lot with the subsequent correlations.

We thank the reviewer for raising this interesting possibility. Indeed there might be a decrease in the retraction fiber diameter around the forming migrasomes. We think however that in our images, the main reason for the reduction of fluorescence near the migrasomes is

a loss of focus resulting from an increase of the height of the retraction fiber during the migrasome growth process.

Regarding the suggestion to use a more general fluorophore, we tried several membrane dyes for retraction fibers and migrasome labelling. However, we found most of the dyes to be toxic to the cells and to abolish the formation of both retraction fibers and migrasomes. In this manuscript we used FM 4-64 for membrane staining which in our hands was the least toxic and the most effective for retraction fibers and migrasomes labelling.

To test whether retraction fibers undergo thinning, we performed AFM imaging of retraction fibers and migrasomes. This approach was the most convenient since AFM scans are not limited by a focal plane and can image varying topography. We did not observe thinning of retraction fibers around migrasomes. We will include these results in other publications as they are beyond the scope of the present article.

4- No supplementary movies 1 and 2 were added, which made it difficult to interpret some of the dynamic results described.

We thank the reviewer for bringing this to our attention. The movies can be now found in the extended data together with four new movies.

5- Lines 66-67: Using all the movies and results already obtained by the authors, would it be possible to characterize the mean time for the event to occur? This quantitative information could be very useful.

We thank the reviewer for this suggestion. We have quantified the mean retention time of FM4-64 positive pre-migrasomes. The results showed that the mean time for pre-migrasomes before TSPAN-GFP recruitment is around 9 min (Figure S2A). Furthermore, we have quantified the mean maturation time of stable migrasomes (FM4-64 and TSPAN4 enriched). The result showed that the mean time is around 112 min (Figure S2B). This is the time it takes TSPAN enrichment in the migrasome to reach steady state.

Figure S2. Mean time of mature migrasome formation. (A) The mean retention time of FM4-64 single-positive pre-migrasomes before Tspan4-GFP recruitment. Data shown represents

the mean (9.2 ± 0.5), error bars are standard error of the mean (SEM), $n= 66$ pre-migrasomes from 10 Tspan4-NRK cells from four independent experiments.(B) The mean maturation time of red-yellow migrasomes (first enriched with FM4-64 and then enriched with FM4-64 and Tspan4-GFP) defined as in Figure 1D. Data shown represents the mean (112 ± 10), error bars are SEM, $n=10$ from 10 TSPAN4-NRK cells from 3 independent experiments.

6- Figure 3: Did the authors perform the OT-micropipette experiment using tension values other than 0.07nN/m? How did the authors select this 0.07nN/m? Is there a threshold value of tension for the event to occur in the biomimetic system?

We have performed the OT-micropipette aspiration experiments at various tensions as written in the Methods section. The range of tensions that we applied on HEK293T-GPMVs is $\Delta\sigma=0.024-0.145$ mN/m. To generate swellings, we performed several tension jumps on each GPMV, yet, we could not correlate any specific feature of the swelling (like the number of swellings, swelling size, stability, etc.) to the magnitude of the tension jump. The value 0.07 mN/m represents the tension jump of the measurements that appear in that particular figure.

Each GPMV had different threshold value of tension (the lowest tension jump) needed for the swelling generation. This might result from variations of the membrane composition and bending rigidity, which can differ for GPMVs from the same cell culture [PMID: 31531398]. The threshold tension we found for swelling generation in HEK-GPMVs was 0.024 mN/m.

Swelling formation in membrane tubes pulled from HEK-GPMVs-TSPAN4-GFP labelled with Dil. A membrane tube was pulled from an aspirated GPMV. Tension jump of $\Delta\sigma=0.024$ mN/m formed swelling on the tube.

Furthermore, we conducted new experiments with GPMVs generated from NRK cells. In these experiments, we had to apply higher tension values ($\Delta\sigma=0.104-0.253$ mN/m) on the GPMVs in order to generate swellings, as compared to GPMVs generated from HEK293T cells. This increase in the threshold tension value might result from higher bending rigidity of NRK cells, which have higher endogenous concentration of tetraspanins than HEK cells (as tetraspanins are known to increase the membrane bending rigidity [PMID: 31371828]). The lowest tension jump, which generated swellings in NRK cells, was 0.104 mN/m.

Swelling formation in membrane tubes pulled from NRK-GPMVs-TSPAN4-GFP labelled with DiI. A membrane tube was pulled from an aspirated GPMV. Tension jump of $\Delta\sigma=0.104$ mN/m formed swelling on the tube.

We further conducted additional control experiments of the tension jump assay on synthetic giant unilamellar vesicles (GUVs) which do not contain any proteins. In these experiments, even very mild tension increase ($\Delta\sigma=0.013$ mN/m) generated swellings. However, we saw a variability in the tension threshold also in the GUV as this tension jump was not able to produce swellings in all the GUVs.

Swelling formation in membrane tubes pulled from GUV. A membrane tube was pulled from an aspirated GUV (DOPC:DOPS:Rhodamine-PE, 90:9.9:0.1%). Tension jump of $\Delta\sigma=0.013$ mN/m formed swelling on the tube.

We added a supplementary Figure S11 and the following text to the manuscript:

“The average threshold of tension jump, which induced periling instability, was different for the two types of GPMVs, as NRK-GPMVs required significantly higher tension jump for swelling formation (Figure S11 and methods). This increase in the threshold tension value might result from a higher bending rigidity of NRK cell membranes, which have higher endogenous concentration of tetraspanins than HEK cells (as tetraspanins are known to increase the membrane bending rigidity¹⁴).”

7- Lines 185-188 and Lines 191-194: the authors used the results for the biomimetic assays to conjecture possible explanations for the phenomenon occurring in cells. I believe it is totally valid but it needs to be much more discussed. For example, the authors never mentioned the cytoskeletal elements present in retraction fibers and how could they participate in the process. It is already known that changes in cell membrane tension are dependent on the cytoskeleton as well as membrane composition. As the results for the biomimetic assays seem to partially resemble the migrasomes but within different time scales (seconds for the GPMV and minutes for the migrasomes), could it be that the presence of cytoskeleton in retraction fibers influenced, somehow, the tension threshold and the time scale? This could be better discussed by the authors and would help to strengthen the mechanism proposed.

We completely agree with the reviewer that the cytoskeleton and the membrane composition are likely to affect the process. In order to emphasize the contribution of cellular factors like the cytoskeleton we now wrote:

“Overall, we investigated the temporal pathway of migrasome formation in conjunction with TSPAN4 dynamics on retraction fibers of live cells and designed a biomimetic system

emulating these processes in a minimal artificial system. In this system we recreated the crucial aspects of migrasome biogenesis observed in live cells: formation of membrane tubes of initial migrasome-like local swellings having relatively low TSPAN concentrations, formation of TSPAN domains and their recruitment to the swellings, dissipation of the swellings in the absence and their stabilization in the presence of TSPAN. The differences between the migrasome formation in cells and the biomimetic system can be related to cellular factors such as the cytoskeleton which may modulate the process in live cells^{5,9,30,31} but are absent from the artificial system. For example, membrane-buckling induced by cytoskeleton was previously implicated in vesicle budding^{32,33}, and regulators of actin dynamics were shown to result in microvesicle formation in tumor cells³⁴. In addition, cortical cytoskeleton may modulate membrane tension^{35,36,37}. Based on the obtained results, it can be concluded that the migrasome formation proceeds in two sequential steps: formation of local swellings on the tubular retraction fibers, and stabilization of these swellings by TSPAN-based membrane domains. Our results suggest that the migrasome biogenesis can be driven by a very limited set of factors: the first step driven by membrane tension with possible involvement of additional factors in live cell membranes, and the second step controlled by specific proteins of the TSPAN family. The particular cellular mechanisms by which membrane tension and TSPAN-based clusters shape the migrasomes, while being partially addressed in the previous work¹⁴, await substantial elaboration. Our results illuminate the mechanism of migrasome biogenesis and demonstrate the crucial role of TSPAN proteins in membrane shaping processes.”

Response to 2nd reviewer:

This paper from the Sorkin lab describes experiments aimed at characterising the cellular mechanisms through which migrasomes – a relatively recently-described category of cell-derived vesicles - bud-off from retraction fibres. The authors have previously shown that a tetraspannin, TSPAN4 is key to migrasome formation and here, they use biophysical approaches to further interrogate the mechanisms through which TSPAN4 and membrane tension conspire to drive migrasome budding. The authors propose a sequential mechanism through which increased membrane tension drives initial migrasome budding from retraction fibres. These buds are then subsequently stabilised – perhaps to facilitate their fission – by recruitment of TSPAN4 to these nascent buds.

In general, the study is elegant and well-executed and provides evidence of a possible mechanism for migrasome genesis. The biophysical approaches are good ones to use to interrogate this process, and the experiments are properly conducted and controlled. Nevertheless, I feel that in order to properly establish that TSPAN4 functions in the way that the authors claim – i.e. to stabilise nascent migrasomes – the authors need to conduct further experiments and controls.

We thank the reviewer for the supportive evaluation of the manuscript. We added additional experiments and controls, which further support our proposal regarding the TSPAN4 function.

Major points:

1. My main point is that the authors need to determine to what extent the mechanism and phenomena that they describe rely on overexpression of TSPAN4 and the extent to which this tetraspannin can mediate stabilisation of migrasome-like buds on retraction fibres when TSPAN4 is present at endogenous levels. The experiments presented in Fig 1E show nicely that there is a recruitment of GFP-TSPAN4 to nascent migrasomes correlates with their stabilisation. The authors also show that migrasome-like structures which form on the membrane tubes pulled from GMPVs made from GFP-TSPAN4 overexpressing cells are 5-times more stable than those formed from GMPVs made from cells that don't overexpress GFP-TSPAN (Fig. 5). However, to establish that such a mechanism could be biologically pertinent, the authors should address the following questions:

a) what is the level of endogenous TSPAN4 in the cells used for this study? Is this insufficient to support migrasome stabilisation?

We thank the reviewer for raising this question. Previously, we analysed the level of endogenous TSPAN4 in NRK cells and found that the transcription level was 427 [PMID: 31371828] meaning that TSPAN4 was the most expressed *TSPAN* gene among TSPAN family.

We have now conducted new experiments with both WT NRK cells and NRK cells overexpressing TSPAN4, see supplementary movies 2, 3 and Figure S4. Our data clearly shows that endogenous levels of TSPAN4 are already sufficient to support stabilisation of migrasomes, while further enrichment of TSPAN4 significantly enhances this stabilisation (Figure S4D). This can be seen from the difference in the percentage of pre-migrasomes that dissipate in the absence of stabilization, Figure S4B), and the increase in the mean number of mature migrasomes per cell (Figure S4C) which changes from 25 to 34.

Figure S4. (A) Time-lapse images of NRK Tspan4-GFP cells and WT NRK cells. Cells were pre-stained by FM4-64, then observed by confocal microscope. Scale

bar, 5 μm . (B) Statistics of percentage of dissipated pre-migrasomes. (C) Statistics of pre-migrasome numbers per cell. (D) Statistics of mature-migrasome numbers per cell. (B),(C),(D): N=54 cells, including 1391 pre-migrasomes for WT NRK group and N=55 cells, including 1899 pre-migrasomes for NRK Tspan4-GFP group from 4 independent experiments.

Further, we have now conducted new biomimetic experiments with GPMVs that were generated from WT NRK and NRK-TSPAN4-GFP cells. Also in the biomimetic system we find that the endogenous level of TSPAN4 is sufficient for stabilization of the swellings, as swellings were significantly longer lived for WT NRK GPMVs compared to HEK293T GPMVs that have relatively low TSPAN concentrations (mean time of 73 and 29 seconds for NRK and HEK293T, respectively, see control groups in the graph below). Moreover, similarly to the results obtained with GPMVs generated from HEK293T cells, swellings that were generated in the presence of TSPAN4-GFP were significantly more stable than swellings in the control group: 314 seconds for TSPAN4-NRK-GPMVs compared to 73 seconds for NRK-GPMVs, see the graph below and Figure S15).

Swelling stability of HEK-GPMVs (black) and NRK-GPMVs (red) with and without TSPAN4

- b) Can the authors perform experiments (similar to those in Figs 1E and Fig. 5) in cell line(s) which endogenously express high levels of TSPAN4? Such as the MGC-803 line used for their previous study.
- c) Is there a correlation between endogenous TSPAN4 expression and swelling stabilisation?
- d) Does knockdown/CRISPR of endogenous TSPAN4 oppose nascent migrasome swelling/stabilisation?

We thank the reviewer for this suggestion. We conducted new experiments with the MGC-803 cell line in which we followed the biogenesis pathway of migrasomes. Next, we created TSPAN4 knockout MGC-803 cells and found that these cells formed migrasomes that were significantly less stable (see supplementary movies 4, 5 and figure S4). Specifically, the percentage of pre-migrasomes which dissipated in the KO cells was higher compared to the dissipated pre-migrasomes in the WT MGC803 cells (Figure S4F). The percentage of mature migrasomes forming from pre-migrasomes was lower for the MGC803-TSPAN4-KO cells

compared to the WT cells (Figure S4G). In addition, the numbers of pre-migrasomes per cell are larger in WT compared to KO cells (figure S4H).

These results clearly show that there is a correlation between the endogenous TSPAN4 expression and the swelling stabilization, and that knockout of endogenous TSPAN4 opposes nascent migrasome stabilization.

Figure S4. (E) Time-lapse images of WT MGC803 cells and MGC803 Tspan4-KO cells. Cells were pre-stained by FM4-64, then observed by confocal microscope. Scale bar, 5 μ m. (F) Statistics of percentage of dissipate pre-migrasomes. (G) Statistics of mature-migrasome numbers per cell. (H) Statistics of pre-migrasome numbers per cell. (F),(G),(H): N=31 cells, including 724 pre-migrasomes for WT MGC803 group and N=31 cells, including 274 pre-migrasomes for MGC803 Tspan4-KO group from 4 independent experiments. All error bars are mean \pm SEM. Two-tailed unpaired t-tests were used for statistical analysis.

Could the authors perform the GPMV/optical tweezers experiments using the wt and TSPAN4 KO MGC-803 lines used in the previous study to investigate the role of endogenous TSPAN4 on migrasome swelling and stabilisation.

Unfortunately, performing GPMV/optical tweezers experiments using WT and TSPAN4 KO MGC-803 cell lines is not possible due to the following reason: In order to perform the OT experiments, the protein of interest must be labelled to assure that the protein is in the GPMV's membrane which is being tested. Even at very high levels of TSPAN4-GFP expression in cells prior to GPMVs generation, only around 10% of the GPMVs contained TSPAN4-GFP. Before we chose a GPMV for the tension jump assay, we scanned each vesicle to see whether it contained TSPAN4-GFP. In WT MGC-803, TSPAN4 is not labelled, hence, we will not be able to know which GPMV contains TSPAN4. As these single-vesicle experiments are very time consuming, it would not be possible to collect data from tens of vesicles to see whether a stabilization effect would be present in a small percentage of the explored vesicles. We note that there is also a technical limitation that prevents us from performing OT experiments with

MGC-803-TSPAN4-KO cells: these cells cannot be shipped from China to Israel due to restrictions imposed by the Chinese government.

Nevertheless, as mentioned above, our new results from the experiments with WT MGC-803 and MGC-803-TSPAN4-KO cells directly show that endogenous levels of TSPAN4 have the same critical role in migrasome stabilization and growth as in the overexpression system.

Furthermore, as mentioned above, we have also conducted new OT experiments with GPMVs generated from WT NRK cells, which have relatively high endogenous concentration of TSPANs. Swellings formed in these experiments were more stable than swellings formed in the WT HEK experiments, with dissipation time ~2.5 times slower for NRK cells compared to HEK cells.

Summarizing, we have conducted OT experiments demonstrating the stabilizing effect of endogenous TSPAN4 on migrasome swellings, as suggested by the reviewer.

I feel that answering these questions is particularly important because the authors' previous study in NCB (2019) has already established that a role for TSPAN4 and biomechanics in migrasome production and used GPMVs – amongst other approaches – to do this. The present manuscript extends these findings in a way that is interesting but, because the present study relies on TSPAN overexpression, the relevance of sequential mechanism proposed here requires a demonstration of whether this can be supported by TSPAN4 when endogenously expressed.

Our new results clearly show that endogenous levels of TSPAN4 support the same mechanism for migrasome formation in which TSPAN4 is not directly involved in the initial swelling of the migrasomes, while it is crucial for migrasome stabilization and growth.

2. In the example shown in Fig. 4, the nascent migrasome forming from the GPMVs appears to form on the tubule at some distance (about 1 micron) from the GPMV. However, the recruitment of GFP-TSPAN4 to the structure seems only occur when it has moved very close (apparently touching) to the GPMV. Does this always occur like this? It would help if the authors could determine this using the gallery of movies that they have collected and determine the distance from the GPMV that initial swelling and GFP-TSPAN4 recruitment respectively occur.

From our observations we cannot determine how many swellings will be generated upon tension increase and in which place along the tether the swellings will occur. The initial swelling and the recruitment of TSPAN4 to the swelling seem to be independent of the distance from the GPMVs. A gallery of images demonstrating that initial swellings are formed at random locations along the tube is now added as a supplementary figure S9.

Figure S9. Initial swelling formation. Confocal microscopy images of a membrane tube pulled from HEK293T-GPMVs containing TSPAN-GFP (green) and dyed with DiIC12 (red). The figure includes 4 pairs of GPMVs in which each pair was subjected to the same tension jump. In the left images the suction pressure was zero (corresponds to zero tension applied, $\sigma = 0$). Next, the tension was increased instantly to the indicated value. It can be seen that the same tension jumps generate different number of swellings in different positions along the tether.

The movement of the swellings is also not consistent as some swellings did not move at all or moved in the opposite direction, away from the GPMV, as can be seen in figure S10.

Figure S10. Swelling movement along a membrane tube. Time-lapse confocal microscopy images of a tube pulled out of a HEK293T-GPMV containing TSPAN4-GFP. After tension induced swelling formation, the swelling was able to move freely along the membrane tube.

3. I do not understand the purpose of the proteomics as conducted in this study? It currently seems to be used to determine that GFP-TSPAN4 is present in GPMVs. I feel that it is more important to determine which TSPANs are 'sorted' into GPMVs and how enriched these are.

TSPAN proteins are found in every cell type [PMID: 19709882]. The proteomics was done in order to estimate the concentration of TSPAN4 in GPMVs before and after overexpression. The results show that GPMVs generated from WT HEK293T cells contained small amount of TSPANs compared to TSPAN4 transfected HEK293T cells. Thus, we can correlate the differences between the two types of GPMVs to the presence of TSPAN4.

4. I do not seem to be able to find the movies for this paper on the NC manuscript tracking site. It is not possible to properly assess these parts of the paper without seeing these movies.

We thank the reviewer for bringing this to our attention. The movies can be now found in the extended data together with four new movies.

Response to 3rd reviewer:

The authors present a series of findings regarding the temporal sequence of events resulting in the formation of migrasomes. While further study of migrasomes is certainly of interest to the EV community, this study would benefit from additional work.

Comments:

1. The experimentation in the manuscript relies on 2 cell lines: 1 used in live cell imaging and another used for their GPMV studies. To gauge the impact of the results a more thorough analysis of key observations across multiple cell types is necessary. The authors themselves refer to this on lines 124/125.

We agree with the reviewer and now add new experiments such that in both live cell imaging and the biomimetic system we demonstrate the key observations in two cell lines per experimental approach. Now we have WT NRK cells, in addition to TSPAN4-GFP NRK cells, as well WT MGC-803 and TSPAN4-KO MGC-803 cells in the live cell imaging experiments. The biomimetic experiments are conducted with HEK-TSPAN4-GFP and WT HEK as well as TSPAN4-GFP NRK and WT NRK. Please see these results in our answers to the second reviewer, comments a-d (Figure S4).

2. The reliance on overexpression studies is concerning, particularly in light of the data presented in Figure 5. How frequent or relevant are supposed migrasomes if in the absence of overexpression their formation is significantly reduced?

The results shown in Figure 5 are meant to show the stabilization effect of TSPAN4 on swellings formed on a membrane tube in a simplified model system. We agree with the reviewer that the stabilization effect of TSPAN4 has to be tested at endogenous levels, and as described above (our answers to reviewer 2 comments a-d) we conducted such experiments with MGC-803 and TSPAN4 KO MGC-803 cells. These additional experiments clearly show the migrasome stabilization effect of endogenous TSPAN4 levels.

Furthermore, we conducted new experiments with NRK-GPMVs which showed that the endogenous level of TSPAN4 is sufficient for stabilization of the swellings, as swellings were significantly longer lived for WT NRK GPMVs compared to HEK GPMVs that have relatively low TSPAN concentrations (mean time of 73 and 29 seconds for NRK and HEK, respectively). Please see these results in our answers to the second reviewer, comment a.

3. Is the formation of migrasomes in their live cell imaging model substrate and/or adhesion dependent? Tetraspanins are known to associate with integrin receptors for example.

We agree that adhesion plays an important role, as TSPAN influences retraction fiber formation by facilitating adhesion via interactions with integrins. These effects have been established and described in previous studies.

Here we report a new finding, namely, that TSPAN4 have a stabilizing effect on migrasomes. We believe that the migrasome stabilization effect of TSPAN4 is independent of adhesion for the following reasons: 1) We observe this in the biomimetic system that lacks adhesion. 2) In live cells, in the absence of TSPAN4 recruitment, pre-migrasomes dissipate and disappear while the retraction fibers stay unchanged. Thus, the two functions of TSPAN4, namely interaction with integrins and stabilization of migrasomes, are independent of each other.

4. The manuscript does not adequately place the work within the broader context of EV biology. How, for example, does their mechanism of formation relate to the previously published works examining the role of hydrostatic pressure in the formation of EVs.

We thank the reviewer for raising these important questions. Our main point, which is based on our recent data, is that migrasomes are not EVs but rather cellular organelles. This is based on our unpublished work that shows that before detaching from cells, migrasomes already carry out important cellular functions as part of cells (under revision). Migrasomes can be seen as a sort of EVs only after detaching from cells. Here we consider the migrasomes before their detachment meaning that they do not belong to EVs.

Independently of the morphological definition, the origin of migrasomes essentially differs from that of EVs. The biogenesis of migrasomes is completely different from formation of EVs such as exosomes, which is the focus of our unpublished studies (under revision in Nature Cell Biology and in Cell Research).

We now add in the introduction as follows:

After formation and maturation, migrasomes separate from the cell body. The detached migrasomes are the new members of the family of extracellular vesicles (EVs), which are important mediators of cell-cell communication¹⁰, as well as spreading of disease^{11,12}, including cancer metastasis¹³. Migrasomes can also potentially be used for diagnostic and therapeutic purposes once their biological roles and formation mechanisms are characterized and better understood.

Are migrasomes released, can they be isolated, are their contents altered with TSPAN overexpression?

After maturation, migrasomes are released from cells and can be isolated (PMID: 31371827, PMID: 31123599, PMID: 32994478). Our unpublished data shows overexpression of tetraspanins does not alter the composition of chemokines enriched in migrasomes.

This said, we realize that without being aware of our recent results, it is straightforward to put migrasomes in the category of EVs. We hope that our upcoming publications will clarify this important issue.

5. What percentage of migrasomes mature? Is TSPAN4 being presented as a definitive marker of migrasome biogenesis, or are other FM labeled vesicular structures also migrasomes simply without TSPAN4?

Please see our response to reviewer 2 for comments a-d (Figure S4). Our new results from WT NRK cell line show that about 33% of pre migrasomes grew to mature migrasomes. In the NRK-TSPAN4-GFP cell line, however, 70% of the pre migrasomes grew to mature migrasomes (Figure S4B). Furthermore, MGC-803 cell line show that about 19% of pre-migrasomes grew to mature migrasomes. In the MGC-803-TSPAN4-KO cell line only about 7% of the pre-migrasomes grew to mature migrasomes. The data show that TSPAN4 promotes the stabilization of pre migrasome and the formation of mature migrasomes.

Yes, TSPAN4 is a definitive marker for migrasome biogenesis for cells expressing TSPAN4. However, there are other tetraspanins which can also promote migrasome formation. In Figure 1, we show there are about 2.2% of migrasomes that are FM positive but TSPAN4 negative, these migrasome are less stable. Our interpretation of this data is that these structures are “premature” migrasomes. These premature migrasome can grow, possibly, driven by other migrasome promoting tetraspanins. However, without the recruitment of TSPAN4-GFP, these structures were mostly unstable, retracted back or collapsed soon after their formation.

6. Are more migrasomes formed in cells that migrate more readily? Is this linked to TSPAN4 expression in those cell types?

Generally speaking, apart from a few exemptions, more migrasomes form in cells that migrate more readily (PMID 35179563). Our previous study has shown that among 33 known tetraspanins, overexpression of 14 of them can promote migrasome formation, thus, beside migration, the capability to form migrasomes is also linked to the expression levels of migrasome promoting tetraspanins.

7. The reader is left confused regarding the Dil-C12 experiments described in line 119. Is this meant to be disordered lipid phase rather than liquid phase? The data seems somewhat contradictory to what the authors propose regarding the role of cholesterol.

Dil-C12 prefers liquid disordered phase (PMID: 17588529). In the initial stage of swellings induced by a tension jump, Dil-C12 is enriched in the swellings, which implies the liquid disordered phase of the swelling membrane. Since we don't have other data to support this, we deleted the following part from line 119: “which is a membrane dye known to preferentially partition in disordered lipid phase.” We do not see a contradiction with the role of cholesterol, as in the stabilization stage, both TSPAN and cholesterol are enriched in the swelling which then is no longer in the liquid disordered phase.

8. The section on TSPAN4 clustering is confusing and should be reworked. The authors utilize the same model as the paper being referenced and yet simultaneously state that the TSPAN4 expression is uniformly distributed but also mobile puncta.

In our previous study [PMID: 31371828] we observed dynamic TSPAN4 clusters on retraction fibers and migrasomes. These previous measurements were done at the stage of stabilization of mature migrasomes. In this study we followed the initial stage of the process preceding the formation and stabilization of mature migrasomes. We noticed that in the initial stage, TSPAN4 is relatively uniformly distributed along the retraction fibers (as seen in Figure 1B). Following the growth of the initial swelling, TSPAN4 formed mobile clusters, which are referred to as mobile puncta, on the retraction fibers. These clusters were recruited to the swelling.

Figure 1B. Time-lapse images of NRK TSPAN4-GFP cells stained by FM4-64. Imaging by structural illumination microscopy (SIM). Scale bar, 5 μm . At this initial stage, TSPAN4-GFP is relatively homogeneously distributed along the retraction fibers.

TSPAN4 domains on retraction fibers following the growth of migrasomes. Scale bar, 5 μm . From [PMID: 31371828]

In our biomimetic system, TSPAN4 were also found to cluster into observable, mobile puncta in the membrane tube, as shown in Figure S12A.

Figure S12A. Time-lapse confocal microscopy images of membrane tube enriched by TSPAN4-GFP at different time points as indicated. The first image ($t=0$) was taken 21 min after pulling the tube from HEK293T-GPMV. GFP puncta, which relates to TSPAN4 domains can be seen on the membrane tube.

We rephrased the section of TSPAN4 clustering in the following way:

As a following step, we examined whether TSPAN4 in our biomimetic system self-organizes into clusters. We found that while being highly enriched in the tubular membranes TSPAN4 can be ununiformly distributed but can also form mobile puncta along the tubes (Fig. S12A). According to the previous observations in cellular retraction fibers¹⁴, these puncta can be classified as clusters. We further demonstrated that TSPAN4 clusters formed also in flat membranes of GPMVs exposed to shear forces, which were induced by buffer flow (Fig. S12B). In this experiment, the GPMVs were injected into a microfluidics chamber under high pressure (1.5 bar), which led to substantial shear forces acting on the vesicle membranes. Altogether our results demonstrate the tendency of TSPAN4 to cluster.

9. Methods are incomplete. Microfluidic work, for example, is not included. There is also no relevant statistical analysis conducted on the data.

We have now added to the methods part a full description of all the protocols.

We have also mentioned explicitly the statistical analysis that we used in each figure caption.

Response to 4th reviewer:

In this paper the authors wanted to explore the formation kinetics and stabilization dynamics of migrasomes; with a focus on tetraspanin 4 (TSPN4). For this study, the authors utilized two systems: live cell confocal microscopy (NKT Cells) and a biomimetic system using giant plasma membrane vesicles (GPMVs). For both experimental investigations, they developed cell-lines over expressing TSPN4. In the biomimetic model, they simulated migrasome-like vesicles generation and used innovative biophysical imaging techniques to evaluate TSPN4

recruitment to membrane swellings that anticipate migrasome formation.

The study seeks to demonstrate that TSPN4 mediates migrasome formation in a two-step process, starting with membrane swelling along retraction fibers and subsequent TSPN4 migration into the swelling site for migrasome growth and stabilization. Their overall conclusion of the two-step formation process is a novel finding. The topic matter is important and investigations such as this are needed to shed light into the mechanisms that leads to the formation of these novel extracellular signaling structures. The authors of this work have led some of the earliest work into migrasomes and their expertise comes across.

The strengths of the paper include novel experimental models that allow for the pursuit of combined optical data alongside dynamic tension changes in their biomimetic system (confocal optical tweezers combined with their model GPMVs). The live cell imaging experiments are more standard, yet the authors do an excellent job in both obtaining excellent images while also translating those images into quantitative data. The use of the GPMVs as a biomimetic model is clever and provides a mechanism (albeit a bit artificial) to explore TSPN4 recruitment dynamics in microsome formation.

The manuscript also had areas of weakness. The conclusions drawn felt overly broad considering that they only using one cell line in each model system, with few replicates. The authors failed at times to help the reader understand some of their experimental decisions (why did they switch the cell line between experiments?), while provided limited background on some of their more innovative methods. As an example, the reasoning behind why the authors chose the focus on TSPN4 is never clearly stated. On line 125 it is hinted it was from previous work from the group, albeit this is not done in a clear way. Overall, this is an interesting piece of work paving the way for the understanding of recently discovered cell signaling structures. With some improvements, this work can have a major impact on the study of environmental cues involved in cellular signaling and communication.

We thank the reviewer for a positive evaluation of the manuscript. We added to the manuscript an Introduction section and elaborated on the technical and interpretational details, which, we believe, will help the readers to better understand the study. We have added experiments with another cell line and performed at least 3 replicates for each measurement. For all the live-cell imaging-based experiments we conducted, we performed at least 3 independent repeats from different plates of cells and across different dates. The migrasomes used for the statistical analysis were randomly selected from all the repeats. We have added this information in our revised manuscript.

Major corrections:

1. While the work is of high interest, it is also weakened by the lack of continuity between the two models. The authors should help to justify why HEK cells were used for the biomimetic model and not the normal rat kidney cells used for the live imaging work. Further, when making conclusions about the overall process of migrasome formation (as the authors do on line 189), these conclusions should be affirmed in more than one cell line (per experimental methods).

The number of replicates for each experiments is also unclear. Currently, their findings are interesting, but only affirmed in this one set up that they have created.

We agree with the reviewer that it is important to affirm our conclusions in more than one cell line per experimental method. Following the reviewer's suggestion, we added additional biomimetic experiments in which we used GPMVs generated from WT NRK cells and NRK cells that overexpress TSPAN4-GFP. The results demonstrated a similar trend where swellings formed on tubes pulled from enriched TSAPN4 GPMVs were more stable than swellings formed in the control group (Figure S14). The results also showed the swellings from the WT NRK cells were more stable than swellings which formed on tubes pulled from GPMVs generated from WT HEK cells (see the graph in our answer to reviewer 2 comment a). We attribute this phenomenon to the higher endogenous expression level of TSAPNs in NRK cells compared to HEK cells.

Figure S14. Swelling stability and TSPAN4 enrichment and tube rupture in NRK-GPMVs. (A) Percentage of tube rupture events with or without TSPAN4 (TSPAN4 n=19 membrane tubes pulled from 9 vesicles, control n=47 membrane tubes pulled from 19 vesicles). (B) Time-lapse confocal microscopy images of swellings on a membrane tube pulled from GPMV containing TSPAN4-GFP (green) and DiI-C12 (red). The time (t) and spring factor (S) are indicated. (C) Confocal microscopy images of swellings on a membrane tube pulled from GPMV containing TSPAN4-GFP (green) after the initial swelling (t=0) and following TSPAN4 swelling recruitment (t=9.9 min). (D) Swellings stability on membrane tubes with or without TSPAN4. In the presence of TSPAN4 in most of the experiments the swellings were stable, and experiments were stopped after 10 min. Hence, the stability of the swelling in the presence of TSPAN4 is probably higher than indicated. TSPAN4 n=25 swellings on 19 membrane tubes pulled from 9 vesicles, Control n=58 swellings on 47 membrane tubes pulled from 19 vesicles. Error bars are Standard error of the mean (SEM).

We added one more cell line also to the live cell imaging experiments. We followed the migrasome biogenesis in MGC-803 cells which endogenously express TSPAN4 and form retraction fibers and migrasomes upon cell migration. Next, we knocked out TSPAN4 from these cells and repeated the experiment. The migrasomes formed by KO cells were less stable. Specifically, the percentage of mature migrasomes which grew from pre-migrasomes was lower for the KO cells compared to the WT ones. This result shows that endogenous TSPAN4 expression stabilizes swellings (see results in our answers to the second reviewer comments a-d).

We added to each figure caption the exact number of replications for each experiment.

2. Their conclusions are also weakened by the limited number of replicates for their experiments. The live cell confocal microscopy conclusions come from four videos. Are they all from the same plating of cells? Are they across different dates? What variations did they include to make sure that these results aren't specific to just one preparation of cells? These details should be provided. Similarly, they should also be provided for the optical tweezers experiments.

We thank the reviewer for pointing out this issue. As for all the living-cell imaging based experiments, we performed at least 3 independent repeats which were from different plates of cells and across different dates. The migrasomes for the statistics were randomly selected from all the repeats. We have added this information in our revised manuscript.

We also added these additional details to the optical tweezer experiments.

3. Limited details are provided on the statistical methods used. Some error bars are shown without reference to what they mean (Fig. 1D). In other times, "SEM" is mentioned, without details.

We thank the reviewer for this issue. We have added the details in Fig. 1D in which the error bars showed the mean \pm SEM (standard error of the mean). We now added the details of the statistical analysis that was used.

4. Hypotheses are made and at times conclusions drawn through reference to their past work (role of cholesterol in TSPAN orientation/formation; lines 153-159) and this falls flat in this work. Data should be collected and details provided in this manuscript should be provided or the authors should refrain from the speculation. For example, could the authors stain cholesterol on these models to see if it associated with TSPAN4?

We thank the reviewer for this suggestion. We have now conducted experiments to measure the cholesterol enrichment level on pre- and mature-migrasomes. The results showed that cholesterol was enriched on pre-migrasomes and the enrichment level was significantly increased on mature migrasomes along with the recruitment of TSPAN4-GFP, which indicates that cholesterol plays a role at both stages of migrasome formation but is more enriched at the second stage.

Figure S15. Cholesterol enrichment in pre and mature migrasomes. (A) Z-stack images of TSPAN4-GFP-expressing cells that were stained using filipin III. Scale bar, 10 μm ; zoom in, 1 μm . (B) Fold change of filipin III intensity on migrasome relative to retraction fiber. Based on series of images as in A, the following parameters were determined: filipin III intensity on migrasome, IpM; filipin III intensity on RF, IpR. Within a pair, the values of IpM/IpR on pre- and mature- migrasomes are displayed. Data shown represents the mean \pm SEM. $n = 76$ pairs of pre-migrasomes, $n = 79$ pairs of mature migrasomes, from 3 independent experiments. Two-tailed unpaired t-tests were used for statistical analysis.

We now write:

We suggest the following explanation for TSPAN4 recruitment to migrasomes. While TSPAN4 molecules exhibit high positive intrinsic curvature corresponding to an effective molecular shape of an inverted cone^{28,29}, TSPAN4 assembly into clusters and larger domains can reduce its intrinsic curvature. This leads to migration of the domains onto the membrane swellings that have a smaller curvature and, therefore, a better curvature compatibility with the domains. The hypothesis of lower intrinsic curvature of large TSPAN-enriched domains compared with single TSPAN proteins is further supported by the finding that TSPANs associate with cholesterol²⁰. Moreover, we quantified the cholesterol enrichment in migrasomes, and found that cholesterol levels increased significantly in mature migrasomes (concomitantly with TSPAN4 enrichment), supporting this hypothesis (Fig. S15). Domains that consist of TSPAN associated with cholesterol and other lipids and proteins should have an

intrinsic curvature substantially lower than that of individual TSPAN molecules, the latter characterized by particularly high values of the intrinsic curvature²⁸.

5. To access and understand both the biology and technology, the reader is required to do a significant background reading. While brevity is always needed, additional details need to be provided. The introduction could have additional background on the biology of migrasomes formation (what is known / not known), while also explaining what previous work led to their choice of TSPAN4 and to support their choice of these particular experiments.

We added an Introduction section to the manuscript elaborating on additional details.

6. The formatting for this work appears off. There is no abstract provided. There is no Discussion. Overall, the manuscript lacks structure. It reads as though it was prepared for another journal, but then sent to Nature Communications.

We reorganized the manuscript.

7. Lines: 85-86 – Relating to Fig 1E, needs a graph showing the TSPAN4 signal quantifying these visible findings.

We thank the reviewer for this suggestion. We have measured the TSPAN4-GFP intensity for the two types of migrasomes, which further indicated that the recruitment of TSPAN4 could stabilize and facilitate the second stage of migrasome formation.

Figure S3. TSPAN4 fluorescence in growing vs shrinking migrasomes. Quantification of TSPAN4-GFP intensity on growing migrasomes or migrasomes that form and shrink back in figure 1E indicated by yellow or white arrows respectively. Error bars are SEM.

8. Related to not providing enough detail for how this work contributes to the field, the references are lacking with just 17 provided. While the field is new, this feels like a lot of previous work was left unreferenced. Charrin et al (2003) work about the interactions of tetraspanins with cholesterol was never mentioned and could support some claims made in the manuscript (p.e).

We added more references to the manuscript which provide more background to the study.

Minor corrections:

Methods overall: Important details are missing from the methods section and one cannot replicate the work as presented. What type of fibronectin is used (human, bovine?). What type of serum? Suppliers are frequently not stated and sometimes they are stated without the region/country, but further in the text this information is provided. Needs to be consistent. Every reagent should at least have a supplier name and then should be written as supplier name/region/country. Suppliers name are the minimum information required.

We add additional details to the methods section.

Some specific notes:

207 - Suppliers for cell lines. added

230 - should come at 211 – Easier to know how cells that are being imaged were made.

Changed

212 - Density? Supplier for fibronectin FM-4 64? added

226 - How many frames? added (60 frames in each movie)

233 - Density? added

235 - Empty vector control? The controls were done on GPMVs that did not contain TSPAN4-GFP. We added an explanation to the methods.

238 - for most of experiments? What about the others? In each figure caption we mentioned whether the GPMVs were labelled with Dil-C12 (the experiments that appear in the manuscript in which we did not label the membrane with Dil-C12 are in figure S10, S12). We conducted the tension jump assay on GPMVs that contained TSPAN4 without Dil-C12 to see whether Dil-C12 affect the process, but we did not observe any difference in the results from GPMVs that contained both TSPAN4 and Dil-C12.

247 - brought? I don't understand what this means. We rephrased the sentence.

251-256 - sentence is too long. Proteins "were" reduced? We rephrased the sentence.

260-261 - wording confusing. "Linear gradients of 5% to 28% for 60 minutes, 28% to 95% for 15 minutes..." We rephrased the sentence.

262 - Thermo is not a company. Thermo Fisher Scientific. We changed accordingly.

264 - MS – Full name followed by abbreviation is missing in text – mass spectrometry (MS). Abbreviations need to be consistent throughout the manuscript and be next to the first time a word is mentioned. We changed this as the reviewer suggested.

305 – LUMICKS was referenced before without country. Be consistent. We now changed this.

Notes on figures

Figure 1 – This reviewer suggests that the reader would benefit from having 'C' and 'D' side by side below figure B. Also, C and D are lacking statistical reference. We rearranged the figure and added statistical reference to D (C shows the fluorescence intensity of specific migrasome).

47- indicated in B by a white arrow. We fixed the sentence

48 - kinds=groups? Kinds sounds non specific. We changed to groups

Figure 2 B - why not fluorescent as well? In cell culture, GPMVs are seen most clearly in phase contrast microscopy.

150 - Published, under review, in preparation? Ref? We added the reference.

151 - This needs a reference. I think they are referencing their previous work (1) but something needs to be referenced here. We rephrased this sentence and added a reference.

Figure 4. B – No stats. Figure 4B shows the fluorescence intensity of figure 4A (i.e., a specific experiment.)

Figure 5. Maybe label A and B makes it easier to follow? We added labels as suggested

Supplementary figures

- Inconsistent TSPAN4 nomenclature. We fixed the TSPAN4 nomenclature

- Figure 2 – lest=left. fixed

- Needs attention to the legends. Done

REVIEWERS' COMMENTS

Reviewer #1 (Remarks to the Author):

The authors have addressed all of my comments and concerns raised in the first draft. I strongly recommend publishing the manuscript in its current format.

Reviewer #3 (Remarks to the Author):

The authors have addressed the comments in my earlier review with data or reasonable explanations. However, the issue of whether migrasomes are EVs, and especially relative to the larger EVs such as oncosomes, needs to be better explained in the text. If their contention, as they state in the rebuttal, is that migrasomes are not EVs but rather cellular organelles, then they need to be more explicit about this in the discussion. They need not be classified as EVs if they are not EVs.

Reviewer #4 (Remarks to the Author):

The authors did an excellent job addressing our critique as well as those from the other reviewers. I believe that the manuscript is significantly strengthened by these edits and I support the publication without further revisions.